# Conformational change within the extracellular domain of B cell receptor in B cell activation upon antigen binding

Zhixun Shen[1†], Sichen Liu[1†], Xinxin Li[1†], Zhengpeng Wan[1], Youxiang Mao[1], Chunlai Chen[2], Wanli Liu[1]*

[1]Laboratory of Lymphocyte Signaling & Molecular Imaging, MOE Key Laboratory of Protein Sciences, School of Life Sciences, Collaborative Innovation Center for Diagnosis and Treatment of Infectious Diseases, Institute for Immunology, Center for Life Sciences, Beijing Key Lab for Immunological Research on Chronic Diseases, Tsinghua University, Beijing, China; [2]School of Life Sciences, Tsinghua-Peking Joint Center for Life Sciences, Beijing Advanced Innovation Center for Structural Biology, Beijing Frontier Research Center for Biological Structure, Tsinghua University, Beijing, China

**Abstract** B lymphocytes use B cell receptors (BCRs) to recognize antigens. It is still not clear how BCR transduces antigen-specific physical signals upon binding across cell membrane for the conversion to chemical signals, triggering downstream signaling cascades. It is hypothesized that through a series of conformational changes within BCR, antigen engagement in the extracellular domain of BCR is transduced to its intracellular domain. By combining site-specific labeling methodology and FRET-based assay, we monitored conformational changes in the extracellular domains within BCR upon antigen engagement. Conformational changes within heavy chain of membrane-bound immunoglobulin (mIg), as well as conformational changes in the spatial relationship between mIg and Igβ were observed. These conformational changes were correlated with the strength of BCR activation and were distinct in IgM- and IgG-BCR. These findings provide molecular mechanisms to explain the fundamental aspects of BCR activation and a framework to investigate ligand-induced molecular events in immune receptors.

DOI: https://doi.org/10.7554/eLife.42271.001

**\*For correspondence:**
liulab@tsinghua.edu.cn

[†]These authors contributed equally to this work

**Competing interests:** The authors declare that no competing interests exist.

## Introduction

Recognition of antigen by B cell receptor (BCR) initiates B cell activation, which ultimately lead to the production of protective antibodies against pathogens (*Kurosaki et al., 2010*). BCR complex comprises a membrane-bound immunoglobulin (mIg) and a noncovalently linked heterodimer composed of Igα and Igβ in 1:1 stoichiometry of mIg: Igα/β (*Tolar et al., 2005*; *Hombach et al., 1990*). Antigen engagement induces phosphorylation of immunoreceptor tyrosine-based activation motifs (ITAMs) in Igα/β by the Src family kinase Lyn, resulting in the triggering of signaling cascades (*Pierce and Liu, 2010*). However, remain unclear is the molecular mechanism through which BCR extracellular antigen binding signal is transmitted across the membrane to the BCR intracellular ITAMs for the purpose of B cell activation. 'Conformational change model' has been proposed to explain the initiation of B cell activation, in which it is supposed that through a series of conformational changes within the BCR complex, the interaction of antigen with the extracellular domain of mIg is transduced to the intracellular domain of BCR (*Harwood and Batista, 2010*).

Although the lack of structural-based evidence significantly limit our understanding of the conformational information of BCR extracellular domains during the transmembrane initiation of BCR

activation, several previous studies support this conformational change model by investigating fluorescence resonance energy transfer (FRET) within BCR cytoplasmic domains (*Tolar et al., 2005*), FRET between plasma membrane and Igα cytoplasmic domain (*Lee and Tolar, 2013*), FRET between membrane and cytoplasmic domain of mIgG (*Chen et al., 2015*) upon antigen engagement, respectively. In these studies, binding of antigen led to a conformational change in the BCR cytoplasmic domains from a closed to an open form (*Tolar et al., 2005*), an increased distance between membrane and Igα but not Igβ (*Lee and Tolar, 2013*), and dissociation of mIgG cytoplasmic tail from cell membrane (*Chen et al., 2015*). In addition to these studies focusing on the conformational changes of the cytoplasmic domains of BCR, it is also reported that the Cμ4 portion in mIgM (and Cγ3 portion in mIgG) of the extracellular domain of BCR is both required and sufficient for antigen-binding induced BCR oligomerization and signaling, suggesting antigen engagement triggered the Cμ4 domain of IgM-BCR (or Cγ3 domain of IgG-BCR) into an orientation in which BCRs are accessible for oligomerization (*Tolar et al., 2009*). Moreover, in our early studies, using a double strand DNA-based tension gauge tether (TGT) experimental system with defined single molecular forces between BCR and surface-immobilized antigen, we observed that IgM- and IgG-BCR exhibited distinct mechanical force sensitivity during activation. IgM-BCR activation was dependent on mechanical force and exhibited a multi-threshold dependence. In contrast, the activation of IgG-BCR only required a low threshold of less than 12 pN (*Wan et al., 2015*; *Wan et al., 2018*; *Wang and Ha, 2013*). Based on the finding that BCR activation was dependent on mechanical forces, it is reasonable to hypothesize that mechanical force delivered by antigen engagement may induce a potential conformational change within BCR complex, which in turn can trigger the transmission of the physical signal outside of the plasma membrane to chemical signal inside of the membrane. Last but not least, antigen binding might induce conformational change of BCR through modulations in the microenvironment (such as cytoskeleton [*Mattila et al., 2013*] or lipid bilayer [*Sohn et al., 2006*]) or altering the charge-charge interactions within BCR complex.

All these indicated the importance to address a long standing question in antigen receptor biology: How the extracellular antigen binding signal at the variable region of mIg is transduced to the intracellular ITAMs at the cytoplasmic domain of Igα/β within BCR complex. In detail, whether or not the conformational change occurs within the extracellular domains of BCR upon antigen binding? If yes, where is the exact location of such conformational change; is it at the region within the heavy chain of BCR mIg or between mIg and Igα/β? Whether or not the conformational change within mIg of BCR complex also occurs in soluble Ig molecule upon antigen binding? Whether or not the conformational change of BCR is correlated to the strength of B cell activation? Here, we combined site-specific labeling methodology and FRET-based assay to monitor the conformational changes in the extracellular domains within BCR complex upon antigen binding. Conformational changes within mIg and in the spatial relationship between mIg and Igβ were captured and quantified, and different isotyped BCRs displayed distinct conformational change patterns upon antigen binding. Meanwhile, these antigen-binding induced conformational changes in IgM-BCR on cell membrane were not observed in the case of soluble IgM monomer antibodies. In addition, the correlation between conformational change and the strength of BCR activation was analyzed. All these results may provide molecular explanations for fundamental aspects of BCR activation during transmembrane signaling transduction of B cells. Moreover, experimental systems developed in the present study may also provide a framework from which to examine ligand-induced intramolecular or intermolecular events in BCR and other immune receptors.

## Results

### Site-specific labeling in mIg heavy chain of BCR complex

To investigate the potential conformational change within extracellular domains of mIg molecule upon BCR engagement with antigen, we constructed various BCR complexes with short tags inserted that permit targeted incorporation of predefined fluorophores by enzymatic labeling (*Zhou et al., 2007*) or metal chelation (*Griffin et al., 1998*) #51]. For IgM-BCR, dually tagged VRC01-IgM-BCR (gp120 antigen-specific) (*Wang et al., 2018a*) was generated to measure the intramolecular FRET between N terminus and Cμ2 domain within mIgM. Specifically, the ybbR tag (*Yin et al., 2006*; *Munro et al., 2014*) was introduced into N terminus of IgM heavy chain in VRC01-

BCR to enable the site-specific labeling at the Fab domain of Ig heavy chain. Fc portion labeling was achieved through the tetracysteine tag (*Hoffmann et al., 2010*; *Nuber et al., 2016*) insertion in Cμ2 domain of VRC01-IgM-BCR, as described for the detailed amino acid position in the Materials and methods section (*Figure 1A*). Dually tagged VRC01-IgM-BCR can be successfully expressed and labeled in 293T cells (*Figure 1B*). Similarly, dually tagged VRC01-IgG-BCR containing ybbR tag in N terminus of IgG heavy chain and tetracysteine tag in Cγ2 domain was constructed (*Figure 1—figure supplement 1A,B*).

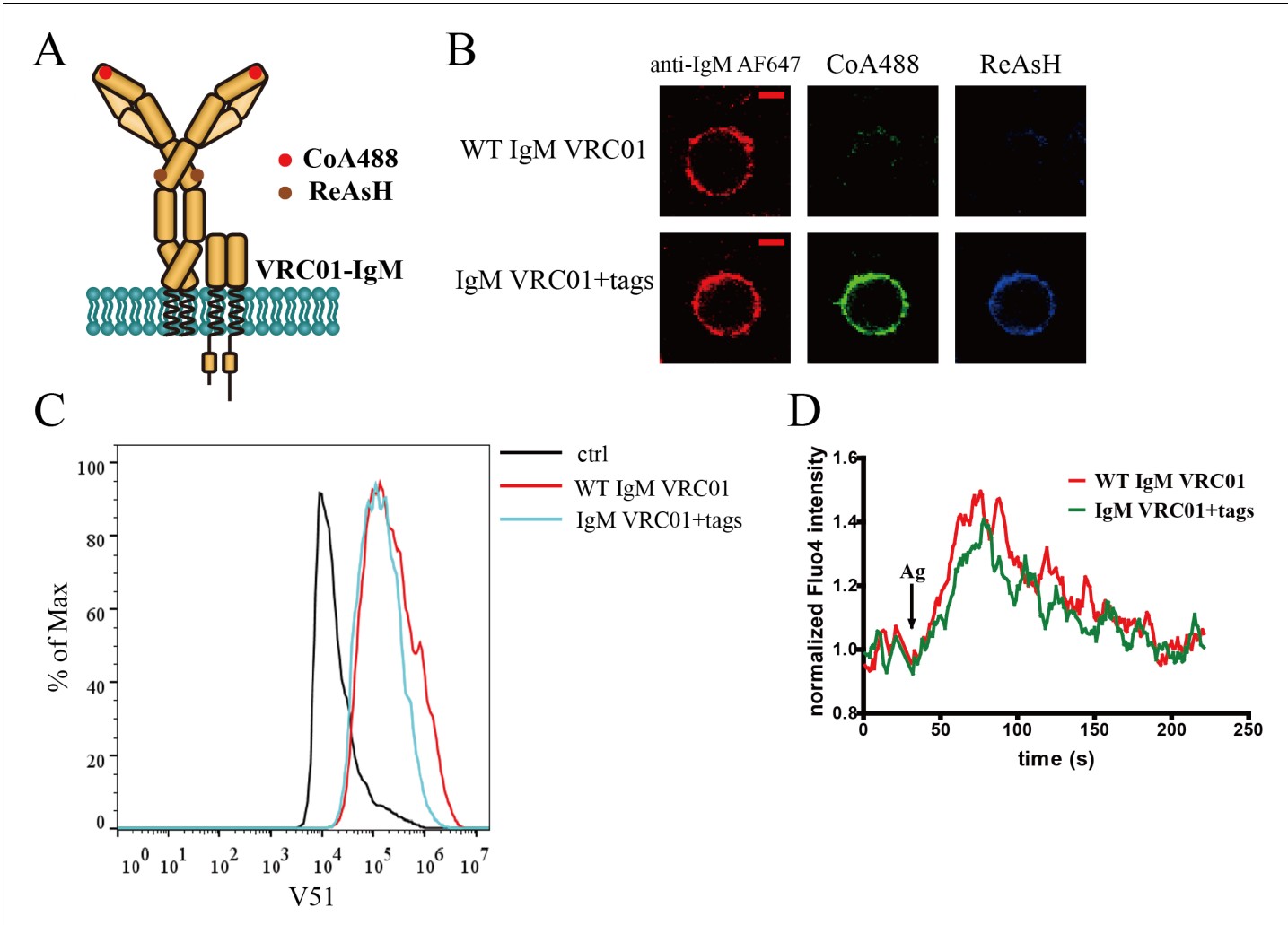

**Figure 1.** Site-specific labeling in mIg heavy chain of IgM-BCR. (**A**) Schematic representation of dually tagged VRC01-IgM-BCR. ybbR tag inserted in N-terminus of IgM heavy chain labeled by CoA 488 was shown as red circle, while tetracysteine tag in Cμ2 region of IgM heavy chain labeled by ReAsH was shown as brown circle. (**B**) Representative confocal images of WT and dually tagged VRC01-IgM-BCR expressed in 293T cells. ybbR tag and tetracysteine tag were stained by CoA 488 and ReAsH, respectively. Alexa Fluor 647 (AF647) Fab fragment of goat anti-human IgM Fc5μ was used for BCR staining. Scale bar, 5 μm. (**C**) WT and dually tagged VRC01-IgM-BCR expressing 293T cells binding with V51 monomer antigen. 293T cells without BCR expression were used as a control. (**D**) $Ca^{2+}$ mobilization analysis of A20II1.6 B cells expressing WT and dually tagged VRC01-IgM-BCR stimulated by V51 trimer antigen.

DOI: https://doi.org/10.7554/eLife.42271.002

The following figure supplements are available for figure 1:

**Figure supplement 1.** Site-specific labeling in mIg heavy chain of IgG-BCR.

DOI: https://doi.org/10.7554/eLife.42271.003

**Figure supplement 2.** Expression of WT and tagged soluble VRC01-IgM monomer.

DOI: https://doi.org/10.7554/eLife.42271.004

Next, we tested whether these modified BCRs expressed in 293T cells or A20II1.6 B cells tolerated the insertion of these short peptide tags. Dually tagged BCRs were expressed in 293T cells for antigen-binding test, while A20II1.6 B cells expressing dually tagged BCRs were used for testing antigen-induced calcium influx. Dually tagged VRC01-IgM-BCR can bind with HIV-1 gp120 antigen (V51 monomer; *Wang et al., 2018a*) as WT VRC01-IgM-BCR (*Figure 1C*), and the stimulation by V51 trimer (A foldon trimerization domain of T4 fibritin was fused to the C terminus of V51 for the production of V51 trimer) (*Wang et al., 2018a*) induced similar calcium influx signal in untagged and dually tagged VRC01-IgM-BCR expressing A20II1.6 B cells (*Figure 1D*). Similarly, dually tagged VRC01-IgG-BCR expressed in 293T cells or A20II1.6 B cells exhibited comparable function as WT VRC01-IgG-BCR (*Figure 1—figure supplement 1C,D*). The oligomeric states of V51 trimer and V51 monomer used were determined by blue native-polyacrylamide gel electrophoresis (BN-PAGE) (*Figure 1—figure supplement 1E*) and confirmed by antigen-induced calcium influx measurement (*Figure 1—figure supplement 1F*). All these results showed the uncompromised function of dually tagged BCRs to recognize antigen and initiate B cell activation.

## Conformational change within the mIgM heavy chain of IgM-BCR upon antigen engagement

Conformational change within mIg heavy chain of antigen-activated VRC01-IgM-BCR was firstly investigated via intramolecular FRET between CoA 488 (CoA-conjugated ATTO 488) label at the N terminus and ReAsH (resorufin arsenical hairpin binder) label in Cμ2 domain (*Figure 2A*). To exclude the interference of intracellular signaling molecules and prevent competition between exogenous and endogenous BCRs in B cells, FRET experiments were firstly performed in 293T cells. 293T cells expressing dually tagged VRC01-IgM-BCR were stimulated by V51 monomer antigen presented by planar lipid bilayers (*Figure 2B*). The results showed that the binding of antigen lowered the FRET efficiency between N terminus and Cμ2 domain as measured by the quantification of donor recovery after acceptor bleaching using total internal reflection fluorescence microscopy (TIRFM) (*Figure 2C, D*). The reduction of FRET signal indicated an increased spatial distance between these two sites upon antigen engagement. To exclude the potential effects from variations in the concentration of donor and acceptor fluorophores, only cells with comparable donor intensity and acceptor intensity were used for analysis in the TIRFM-based acceptor photobleaching FRET experiments (*Figure 2D*). As a further validation, fluorescence lifetime imaging microscopy FRET (FLIM-FRET) measurement was also performed as it can be applied to determine FRET efficiency irrespectively of the concentration of donor and acceptor fluorophores. Decreased FRET between these two sites within IgM heavy chain resulted from V51 monomer antigen engagement was similarly obtained, as indicated by the increased fluorescence lifetime of CoA 488 (donor chromophore) and thus reduced FLIM-FRET signal after antigen binding (*Figure 2E–H*). We also repeated the same FRET measurement via acceptor photobleaching method in A20II1.6 B cells to confirm that the reduced FRET signal within the ectodomain of BCR also occurred in B cells (*Figure 2I*). The increased distance between N terminus and Cμ2 domain of IgM-BCR upon antigen binding suggested that the antigen-induced conformational change might originate at antigen binding site.

## Distinguishing intermolecular FRET and intramolecular FRET

In all the above FRET experiments, some of the FRET signals might result from inter-complex juxtaposition of different BCRs since BCR oligomerization will occur upon antigen engagement. To minimize the involvement of FRET between neighboring BCRs, we thus only analyzed a subset of cells that had equivalent mean fluorescent intensity of donor and acceptor (i.e. equal density of dually tagged BCRs) in all experiments (*Figure 2D,I*). To further exclude these potential effects from intermolecular FRET, we similarly performed FRET experiments in 293T cells expressing dually tagged VRC01-IgM-BCR or two types of singly tagged VRC01-IgM-BCR. Plasmids carrying dually tagged VRC01-IgM-BCRs were diluted with plasmids carrying untagged BCRs and expressed in 293T cells to generate only intramolecular FRET within one individual heavy chain of mIg (*Figure 2—figure supplement 1A*), while plasmids carrying two types of singly tagged VRC01-IgM-BCRs (ybbR tag and tetracysteine tag are in different BCRs, respectively) were diluted with untagged BCRs and expressed in 293T cells to serve as a negative control presenting potential intermolecular FRET (*Figure 2—figure supplement 1B*), which might include a fraction of interchain FRET (FRET between

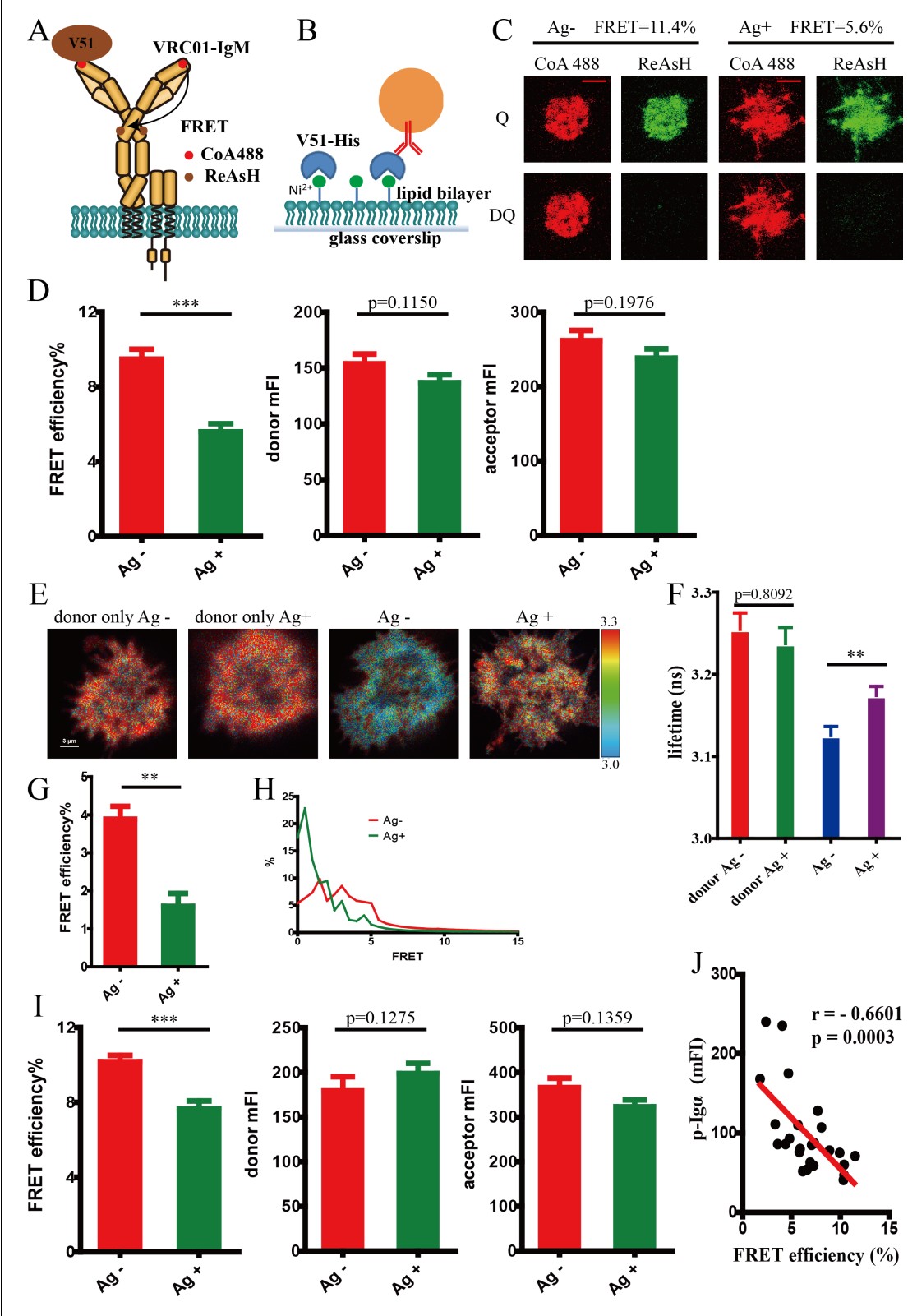

**Figure 2.** Conformational change within mIg heavy chain of VRC01-IgM-BCR upon antigen engagement. (**A**) Schematic illustration showing FRET efficiency between CoA 488 label at N-terminus (red circle) and ReAsH label at Cμ2 domain (brown circle) of dually tagged VRC01-IgM-BCR. (**B**) Schematic illustration showing V51 monomer fused with poly-histidine tag presented on the surface of Ni$^{2+}$-containing planar lipid bilayer (PLB) membranes on supported glass coverslips. (**C–D**) Dequenching FRET to measure the FRET efficiency between CoA 488 and ReAsH in 293T cells

*Figure 2 continued on next page*

*Figure 2 continued*

expressing dually tagged VRC01-IgM-BCR activated by V51 monomer antigen. Representative TIRFM images of V51-activated and non-activated 293T cells were shown. Cells with equal donor intensity and equal acceptor intensity were used for FRET analysis. Scale bar is 5 μm. Data are from at least 34 cells over five independent experiments. Q: quench; DQ: dequench. (E–H) FLIM-FRET to measure the FRET efficiency between CoA 488 and ReAsH of dually tagged VRC01-IgM-BCR expressed in 293T cells stimulated by V51 monomer antigen. (E) Representative FLIM images of V51-activated and non-activated 293T cells stained by donor only (CoA 488) or stained by donor and acceptor (CoA 488 + ReAsH) were shown. Scale bar is 3 μm. (F–G) Fluorescence lifetime and lifetime-based FRET efficiency of dually tagged VRC01-IgM-BCR expressing 293T cells. (H) FRET efficiency distribution of all measured dually tagged VRC01-IgM-BCR molecules. Data are from at least 11 cells. (I) Dequenching FRET to measure the FRET efficiency between CoA 488 and ReAsH in A20II1.6 B cells expressing dually tagged VRC01-IgM-BCR activated by V51 monomer antigen. Cells with equal donor intensity and equal acceptor intensity were used for FRET analysis. Data are from at least 28 cells. (J) The association of FRET efficiency between CoA 488 and ReAsH with the mean intensity of the p-Igα recruited into the immunological synapse in A20II1.6 B cells was indicated by Spearman's rank correlation coefficient (r). Error bars represent mean ± SEM. Two-tailed t-tests were used for the statistical comparisons. ***p<0.001; **p<0.01.

DOI: https://doi.org/10.7554/eLife.42271.005

The following figure supplements are available for figure 2:

**Figure supplement 1.** Conformational change within mIg heavy chain of VRC01-IgM-BCR upon antigen engagement was mainly attributed to the change of intramolecular FRET.

DOI: https://doi.org/10.7554/eLife.42271.006

**Figure supplement 2.** Conformational change within Ig heavy chain of soluble VRC01-IgM monomer upon antigen engagement.

DOI: https://doi.org/10.7554/eLife.42271.007

CoA 488 and ReAsH labeled in two distinct heavy chains within one individual VRC01-IgM-BCR). Under the same experimental condition as above, antigen binding caused significant lower FRET between CoA 488 and ReAsH in cells expressing dually tagged VRC01-IgM-BCRs (*Figure 2—figure supplement 1C*). Meanwhile, FRET signal between CoA 488 and ReAsH in singly tagged VRC01-IgM-BCRs expressing cells was very low, and no FRET change was observed between these two fluorophores upon antigen engagement (*Figure 2—figure supplement 1D*), suggesting the alteration of FRET between N terminus and Cμ2 domain within IgM heavy chain generated by antigen-BCR interaction was mainly attributed by the increase of intramolecular distance between these two sites, rather than the intermolecular distance changes. We also compared the intramolecular FRET and intermolecular FRET between CoA 488 and ReAsH in 293T cells expressing non-diluted tagged VRC01-IgM-BCRs. Dually tagged VRC01-IgM-BCRs without dilution were expressed in 293T cells to generate intramolecular FRET (with a fraction of intermolecular FRET), while two types of singly tagged RC01-IgM-BCRs without dilution were expressed in 293T cells to generate intermolecular FRET. We found that intermolecular FRET accounted for about 1/3 of the detected intramolecular FRET signal, and antigen engagement significantly decreased intramolecular FRET, but not intermolecular FRET (*Figure 2—figure supplement 1E,F*).

## Antigen-binding induced conformational change within mIg heavy chain is closely correlated to the strength of B cell activation

To assess whether antigen-binding induced conformational change within mIgM heavy chain is functionally relevant, we examined the association of the level of synaptic recruitment of phosphorylated Igα (p-Igα) with FRET efficiency between N terminus and Cμ2 domain in dually tagged VRC01-IgM-BCR expressing A20II1.6 B cells upon antigen stimulation. Inverse correlation between the FRET efficiency and the mean fluorescent intensity (mFI) of p-Igα was observed (*Figure 2J*), indicating the spatial distance between N terminus and Cμ2 domain within mIg is closely associated with the strength of B cell activation. However, it should be noted that another possible explanation for this association is that decreased FRET signal could be a consequence of more efficient antigen binding of the cell, thus enhancing the recruitment of p-Igα.

## Conformational change within mIg heavy chain is specific to membrane IgM-BCR

Since the crystal structures of antigen-antibody complex indicate that antigen binding does not transmit conformational changes to the membrane-proximal regions of Ig molecule (*Metzger, 1974*; *Metzger, 1992*), we compared the conformational change within Ig of soluble VRC01-IgM and VRC01-IgM-BCR expressed on cell membrane. We expressed and purified dually tagged soluble

VRC01-IgM monomers (*Figure 1—figure supplement 2A,B*), which can normally bind with gp120 antigen expressed on cells (*Figure 1—figure supplement 2C*) and can be labeled site-specifically (*Figure 1—figure supplement 2D*). Through acceptor photobleaching FRET method, we found no FRET change between N terminus and Cμ2 domain within soluble VRC01-IgM upon antigen binding (*Figure 2—figure supplement 2A–C*). Similarly, using FLIM-FRET method, no change of spatial distance between these two sites in soluble VRC01-IgM after antigen engagement was observed (*Figure 2—figure supplement 2D–G*). Therefore, data generated by different FRET detection methods demonstrated the decreased proximity between N terminus and Cμ2 domain within extracellular domains of IgM-BCR upon antigen stimulation, while no such change of spatial distance occurs between N terminus and Cμ2 domain within soluble IgM.

## IgM- and IgG-BCR exhibited distinct conformational change pattern within the mIg heavy chain upon antigen binding

Given that the activation of the IgM- and IgG-BCR exhibited different mechanical force sensitivity (*Wan et al., 2015*), which might be correlated to the different conformational change within the mIg heavy chain, we thus investigated the antigen binding-induced FRET changes in the mIg heavy chain of IgM- versus IgG-BCR (*Figure 3A*). In comparison with IgM-BCR, distance between N terminus and Cγ2 domain in VRC01-IgG-BCR remained unchanged after interacting with V51 monomer antigen, as illustrated by the unaltered FRET between CoA 488 label at the N terminus and ReAsH label at Cγ2 of IgG heavy chain (*Figure 3B*, *Figure 3—figure supplement 1A,B*).

In addition, our previous study showed that the low mechanical force threshold of IgG-BCR activation is dependent on its cytoplasmic tail. Mechanistically, IgG cytoplasmic tail lowers the mechanical force threshold of IgG-BCR activation by enriching phosphatidylinositol (4,5)-biphosphate (PI(4,5) P2) into IgG-BCR membrane microdomains and potentially interacting with plasma membrane through its positively charged residues (*Wan et al., 2015*; *Wan et al., 2018*). This finding drove us to speculate that mIg cytoplasmic tail might be responsible for the distinct conformational change within the mIg heavy chain between IgM- and IgG-BCR in response to antigen binding. To determine the effect of mIg cytoplasmic tail on conformational change within mIg, we constructed VRC01-IgM-BCR equipped with a mIgG cytoplasmic tail (termed MMG-BCR thereafter) and VRC01-IgG-BCR equipped with a mIgM cytoplasmic tail (termed GGM-BCR thereafter), both of which possess one ybbR tag at N terminus of mIg heavy chain as VRC01-IgM- and VRC01-IgG-BCR (*Figure 3A*). In all constructs, FRET acceptor chromophore was ReAsH labeled at Cμ2 or Cγ2 region of mIg. The results showed that GGM-BCR exhibited the similar FRET change after antigen stimulation as IgM-BCR. In contrast, no significant conformational change occurred in antigen-stimulated MMG-BCR, resembling the behavior of IgG-BCR (*Figure 3B*, *Figure 3—figure supplement 1A,B*). In addition, we found that IgG cytoplasmic tail slightly decreased the basal FRET efficiency between CoA 488 and ReAsH in VRC01-BCR without antigen-binding (*Figure 3—figure supplement 1C,D*), while the difference was not significant. Moreover, two-way ANOVA showed a significant interaction effect between Ig cytoplasmic tail and antigen engagement on FRET between N terminus and Cμ2 (or Cγ2), indicating that Ig cytoplasmic tail significantly affected the antigen-induced conformational change within mIg of BCR. Two-way ANOVA with Sidak's correction for multiple comparisons confirmed that antigen-binding significantly increased distance between N terminus and Cμ2 (or Cγ2) within mIg only in the absence of IgG cytoplasmic tail, while the presence of IgG cytoplasmic tail did not significantly increase the basal distance between these two sites of BCR without antigen-binding (*Figure 3—figure supplement 2*). All these results suggested that IgM- and IgG-BCR exhibit distinct conformational change within the mIg heavy chain upon antigen binding and such difference is dependent on the cytoplasmic tail of mIg.

## Site-specific labeling of the extracellular domain of Igβ in BCR complex

It is assumed that the antigen engagement of mIg induces a conformational change within BCR complex to initiate recruitment of Lyn and phosphorylation of ITAMs at the cytoplasmic domain of Igα/β, facilitating the assembly of the membrane proximal signalosome in B cell. Thus, it is an intriguing hypothesis that the antigen-induced conformational change within mIg shall be somehow transmitted to Igα/β heterodimer, causing a further conformational change in the spatial relationship between mIg molecule and Igα/β within the BCR complex.

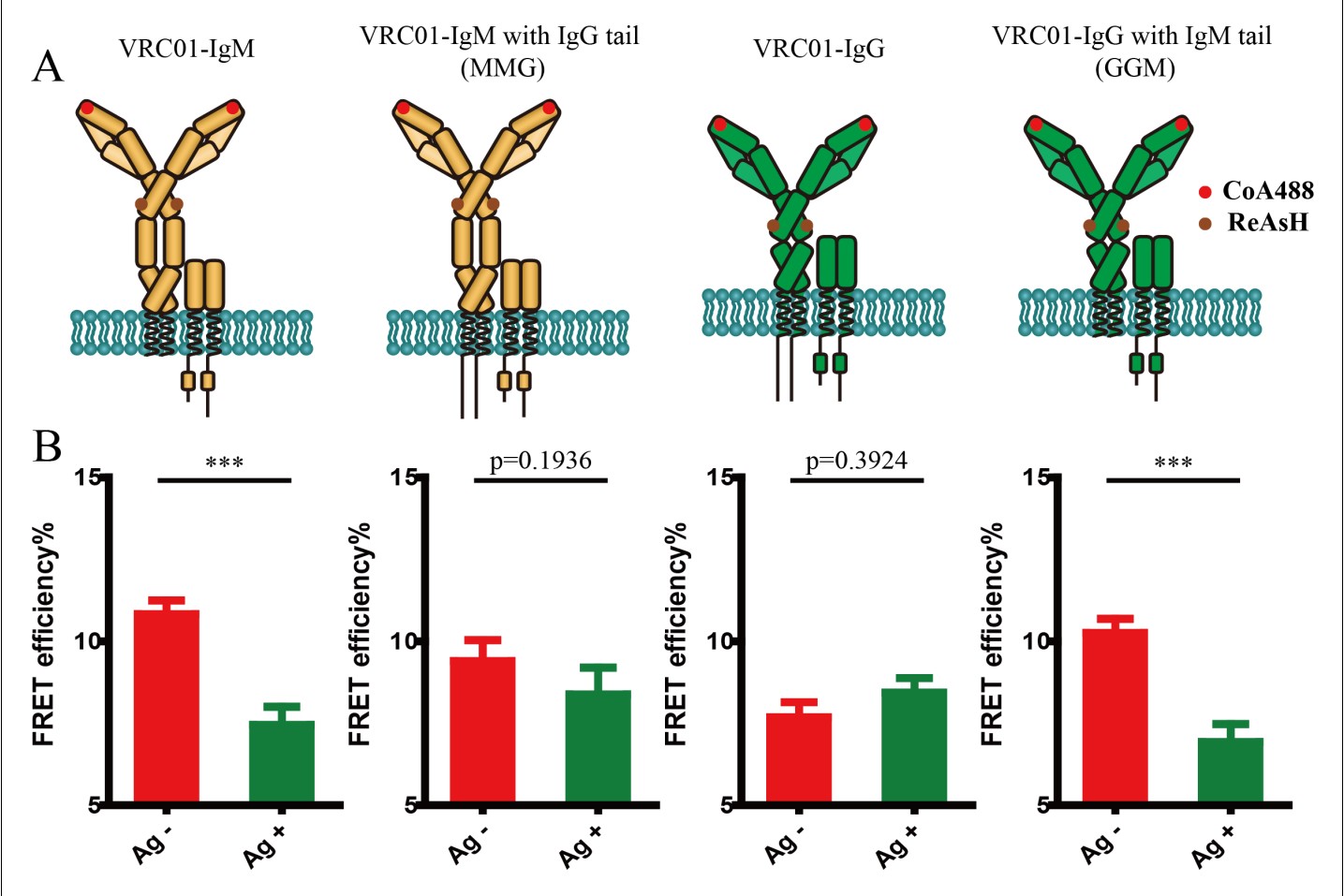

**Figure 3.** Conformational change within mIg heavy chain is dependent on the cytoplasmic tail. (**A**) Schematic illustration of different types of VRC01-BCR equipped with different cytoplasmic tail. FRET efficiency between CoA 488 label at N-terminus (red circle) and ReAsH at Cμ2 domain (or Cγ2 portion, brown circle) of dually tagged VRC01-BCR was examined. (**B**) Dequenching FRET to measure the FRET efficiency between CoA 488 and ReAsH in 293T cells expressing dually tagged VRC01-BCR activated by V51 monomer antigen. From left to right: IgM, MMG, IgG, GGM. Error bars represent mean ± SEM. Data are from at least 19 cells over two independent experiments. Two-tailed t-tests were used for the statistical comparisons. ***p<0.001.

DOI: https://doi.org/10.7554/eLife.42271.008

The following figure supplements are available for figure 3:

**Figure supplement 1.** Conformational change within mIg heavy chain is dependent on the cytoplasmic tail.
DOI: https://doi.org/10.7554/eLife.42271.009

**Figure supplement 2.** Conformational change within mIg heavy chain is dependent on the cytoplasmic tail.
DOI: https://doi.org/10.7554/eLife.42271.010

For the detection of intermolecular FRET between mIg and Igα/β heterodimer, we firstly tried to site-specifically label Fc region of mIg or Igα/β by ybbR tag insertion but failed. The difficulty of labeling might be caused by the potential steric hindrance effect between mIg and Igα/β (*Radaev et al., 2010*). Therefore, to measure FRET between mIg and Igα/β, tetracysteine tag was introduced into the extracellular domain of Igβ in both IgM- and IgG-BCR, and Fab fragment of antibody was used to label the Fc portion of mIg heavy chain (*Figure 4—figure supplement 1A*, *Figure 4—figure supplement 2A*) for measurement of FRET between mIg and Igβ. Confocal microscopy of transfected 293T cells showed that VRC01-BCRs with tetracysteine tag in Igβ were expressed and site-specifically labeled (*Figure 4—figure supplement 1B*, *Figure 4—figure supplement 2B*). Both WT and tagged VRC01-BCR can bind with V51 monomer (*Figure 4—figure supplement 1C*, *Figure 4—figure supplement 2C*). V51 trimer can induced calcium influx in tagged

VRC01-BCR expressing A20II1.6 B cells, although the calcium signals in tagged VRC01-IgM-BCR expressing A20II1.6 B cells showed a little less sustained signal than cells expressing untagged VRC01-IgM-BCR (*Figure 4—figure supplement 1D*, *Figure 4—figure supplement 2D*). All these results indicated that VRC01-BCR tolerates the tetracysteine tag inserted in Igβ.

## IgM- and IgG-BCR exhibited distinct conformational change pattern in the spatial relationship between mIg and Igβ upon antigen binding

We measured FRET efficiency between Alexa Fluor 647 label at Fc5μ fragment within IgM heavy chain and ReAsH label in Igβ of VRC01-IgM-BCR in 293T cells (*Figure 4—figure supplement 3A*). The result showed that no FRET change between these two sites occurred upon antigen binding, suggesting no change of spatial distance occurred between mIg and Igβ in IgM-BCR complex (*Figure 4—figure supplement 3B,C*).

Considering that IgM- and IgG-BCR showed distinct conformational change within mIg heavy chain after antigen engagement as aforementioned, we also examined the distance change between mIg and Igβ in antigen-activated VRC01-IgG-BCR (*Figure 4A*). We found a decreased intermolecular FRET efficiency between Alexa Fluor 647 label at Fcγ fragment of IgG heavy chain and ReAsH label in extracellular domain of Igβ measured by acceptor photobleaching FRET method in 293T cells (*Figure 4B–C*), indicating an increased spatial distance between mIg heavy chain and Igβ in VRC01-IgG-BCR upon the stimulation by membrane antigen. Similarly, cells with equal donor intensity and acceptor intensity were analyzed (*Figure 4C*). As a further validation, FLIM-FRET measurement confirmed that membrane V51 monomer antigen similarly caused a FRET reduction between these two sites in VRC01-IgG-BCR (*Figure 4D–G*). Increased distance between mIg heavy chain and Igβ was also confirmed in tagged VRC01-IgG-BCR expressing A20II1.6 B cells after binding with antigen (*Figure 4H*). Moreover, FRET efficiency between these two sites was found to be inversely correlated with the level of synaptic recruitment of p-Igα (*Figure 4I*), indicating the spatial distance between extracellular domains of mIg and Igβ is closely associated with the strength of B cell activation.

Since the cytoplasmic tail of mIg molecule was found to be involved with the mechanical force requirement of BCR activation (*Wan et al., 2015*; *Wan et al., 2018*) and was related with the conformational change within extracellular domains of mIg in BCR complex upon antigen engagement, as described in our published study and abovementioned result, respectively, we thus investigated the effect of mIg cytoplasmic tail on antigen-induced conformational change in the spatial relationship between mIg molecule and Igβ (*Figure 5A*). In line with the change observed in antigen-stimulated VRC01-IgG-BCR, spatial distance between mIg heavy chain and Igβ in IgG cytoplasmic tail carrying VRC01-MMG-BCR increased upon antigen engagement, as illustrated by the decreased FRET efficiency. By contrast, VRC01-GGM-BCR possessing IgM cytoplasmic tail resembled the behavior of VRC01-IgM-BCR, in which the distance between mIg and Igβ remained unaffected after antigen activation (*Figure 5B*, *Figure 5—figure supplement 1A,B*). In addition, we observed that the presence of IgG cytoplasmic tail significantly increased the basal FRET efficiency between ReAsH and AF647 in VRC01-BCR without antigen binding (*Figure 5—figure supplement 1C,D*). Moreover, two-way ANOVA showed a significant interaction effect between Ig cytoplasmic tail and antigen engagement on FRET between mIg and Igβ, indicating that Ig cytoplasmic tail significantly affected the antigen-induced conformational change in the spatial relationship between mIg and Igβ. Two-way ANOVA with Sidak's correction for multiple comparisons confirmed that antigen-binding significantly increased the distance between mIg and Igβ only in the presence of IgG cytoplasmic tail, while IgG cytoplasmic tail significantly decreased the basal distance between these two molecules (*Figure 5—figure supplement 2*).

In summary, these results suggested that the antigen-induced conformational change within mIg heavy chain might be transmitted to Igα/β, decreasing proximity between mIg and Igβ of IgG-BCR. In addition, conformational change in the spatial relationship between mIg and Igβ is dependent on the cytoplasmic tail of mIg and is associated with the initial intracellular signaling in activated B cell.

## Discussion

B cell activation is triggered by BCR-antigen engagement. When BCRs encounter antigens, the mobility of BCRs will be reduced followed by BCR oligomerization and the formation of BCR microclusters (*Tolar et al., 2005*; *Harwood and Batista, 2010*; *Mattila et al., 2013*; *Liu et al., 2010a*;

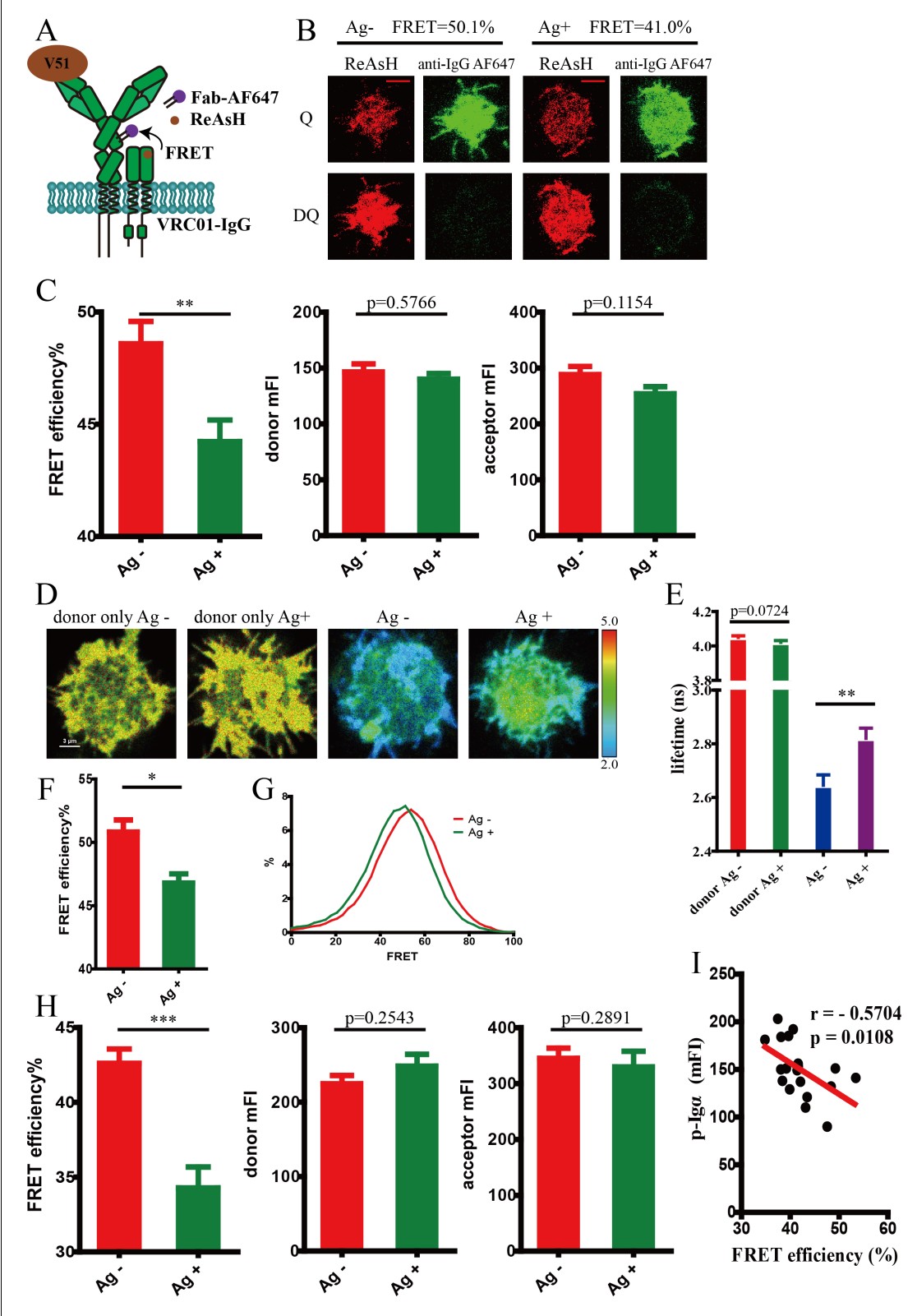

**Figure 4.** Conformational change in the spatial relationship between mIg heavy chain and Igβ in VRC01-IgG-BCR upon antigen engagement. (**A**) Schematic illustration showing FRET efficiency between ReAsH label in Igβ (brown circle) and AF647 label in Fc region (purple circle) of tagged VRC01-IgG-BCR. (**B–C**) Dequenching FRET to measure the FRET efficiency between ReAsH and AF647 in 293T cells expressing tagged VRC01-IgG-BCR activated by V51 monomer antigen. Representative TIRFM images of V51-activated and non-activated 293T cells were shown. Cells with equal donor

*Figure 4 continued on next page*

*Figure 4 continued*

intensity and equal acceptor intensity were used for FRET analysis. Scale bar is 5 µm. Data are from at least 30 cells over two independent experiments. Q: quench; DQ: dequench. (**D–G**) FLIM-FRET to measure the FRET efficiency between ReAsH and AF647 of tagged VRC01-IgG-BCR expressed in 293T cells stimulated by V51 monomer antigen. (**D**) Representative FLIM images of V51-activated and non-activated 293T cells stained by donor only (ReAsH) or stained by donor and acceptor (ReAsH +AF647) were shown. Scale bar is 3 µm. (**E–F**) Fluorescence lifetime and lifetime-based FRET efficiency of tagged VRC01-IgG-BCR expressing 293T cells. (**G**) FRET efficiency distribution of all measured tagged VRC01-IgG-BCR molecules. Data are from at least nine cells. (**H**) Dequenching FRET to measure the FRET efficiency between ReAsH and AF647 in A20II1.6 B cells expressing tagged VRC01-IgG-BCR activated by V51 monomer antigen. Cells with equal donor intensity and equal acceptor intensity were used for FRET analysis. Data are from at least 24 cells. (**I**) The association of FRET efficiency between ReAsH and AF647 with the mean intensity of the p-Igα recruited into the immunological synapse in A20II1.6 B cells was indicated by Spearman's rank correlation coefficient (r). Error bars represent mean ± SEM. Two-tailed t-tests were used for the statistical comparisons. ***$p<0.001$; **$p<0.01$; *$p<0.05$.

DOI: https://doi.org/10.7554/eLife.42271.011

The following figure supplements are available for figure 4:

**Figure supplement 1.** Site-specific labeling in Igβ of IgM-BCR.

DOI: https://doi.org/10.7554/eLife.42271.012

**Figure supplement 2.** Site-specific labeling in Igβ of IgG-BCR.

DOI: https://doi.org/10.7554/eLife.42271.013

**Figure supplement 3.** Conformational change in the spatial relationship between mIg heavy chain and Igβ in VRC01-IgM-BCR upon antigen engagement.

DOI: https://doi.org/10.7554/eLife.42271.014

*Liu et al., 2010b*). BCR is an extraordinary receptor that can efficiently discriminate the biochemical and biophysical features of an antigen, such as the stiffness of the substrates presenting the antigen (*Wan et al., 2013*; *Zeng et al., 2015*; *Shaheen et al., 2017*; *Wang et al., 2018b*) and mechanical force signals delivered by antigen (*Wan et al., 2015*; *Wan et al., 2018*; *Natkanski et al., 2013*; *Nowosad et al., 2016*; *Spillane and Tolar, 2017*). However, it remains an open but fundamental question in terms of the molecular mechanism of how the BCR transduces antigen signals. For example, it is still elusive of how the extracellular antigen binding signal is transduced to the cytoplasmic domain of BCR complex to trigger downstream signaling cascades?

In the field, it has been proposed that the conformational change model of BCR might be applicable in the BCR-mediated initiation of B cell activation, mirroring the similar model of T cell receptor (TCR) activation (*Gil et al., 2002*; *Lee et al., 2015*; *Ma et al., 2008*; *Xu et al., 2008*; *Blanco et al., 2014*; *Swamy et al., 2016*; *Schamel et al., 2017*; *Dustin and Davis, 2014*; *Depoil and Dustin, 2014*; *Dustin, 2009*). However, several crystal structures of antibodies in the presence of soluble antigens indicate that antigen binding seems incapable of propagating significant conformational changes to membrane-proximal regions of Ig (*Davies and Metzger, 1983*; *Davies et al., 1990*; *Metzger, 1978*). Our results in this report also confirmed that no conformational change occurs within soluble VRC01-IgM molecule upon membrane antigen engagement via FRET measurement between N terminus and Cµ2 region. We shall emphasize that there is a significant difference between the conditions in which the characterization of antigen-antibody complexes was examined and those under which we explored the antigen-BCR interaction on cell plasma membrane. Indeed, several studies directly focused on conformational changes within intracellular domains of BCR (*Tolar et al., 2005*; *Lee and Tolar, 2013*; *Chen et al., 2015*) suggest the lack of structural-based evidence at this stage does not preclude the possibility that conformational changes may contribute to the initiation of B cell activation. In addition, according to our published studies, the interaction between cytoplasmic tail and cell membrane is necessary for the antigen-induced conformational change in cytoplasmic domain of BCR (*Chen et al., 2015*). It is also reported that extracellular conformational change-mediated oligomerization of the BCR requires the WTxxST motif in the transmembrane region of mIg (*Tolar et al., 2009*), which is lacked in soluble Ig. These findings indicate that interaction between BCR complex and cell membrane might be crucial for conformational changes within ectodomains of BCR upon antigen engagement. Nevertheless, whether conformational changes occur at extracellular domains of BCR remains unknown. Investigation on this process will facilitate us to understand how BCR transmit extracellular antigen signal across the membrane to the intracellular component to trigger downstream signaling during early B cell activation, providing us molecular mechanism for fundamentals of B cell responses. In the present study,

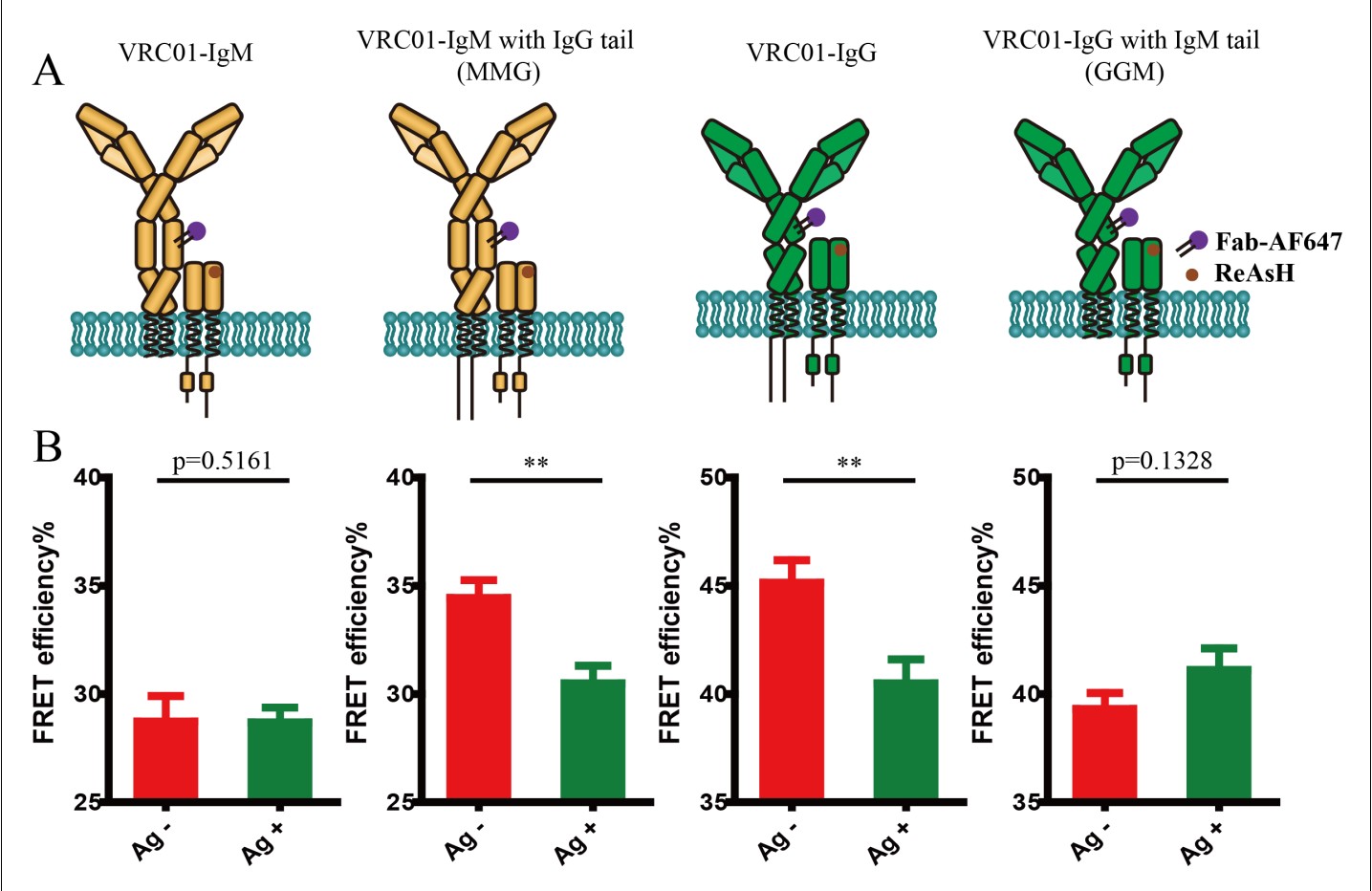

**Figure 5.** Conformational change in the spatial relationship between mIg heavy chain and Igβ is dependent on the cytoplasmic tail. (**A**) Schematic illustration of different types of VRC01-BCR equipped with different cytoplasmic tail. FRET efficiency between ReAsH label at extracellular domain of Igβ (brown circle) and AF647 label at Fc portion (purple circle) of tagged VRC01-BCR was detected. (**B**) Dequenching FRET to measure the FRET efficiency between ReAsH and AF647 in 293T cells expressing tagged VRC01-BCR activated by V51 monomer antigen. From left to right: IgM, MMG, IgG, GGM. Error bars represent mean ± SEM. Data are from at least 19 cells over two independent experiments. Two-tailed t-tests were used for the statistical comparisons. \*\*p<0.01.
DOI: https://doi.org/10.7554/eLife.42271.015

The following figure supplements are available for figure 5:

**Figure supplement 1.** Conformational change in the spatial relationship between mIg heavy chain and Igβ is dependent on the cytoplasmic tail.
DOI: https://doi.org/10.7554/eLife.42271.016

**Figure supplement 2.** Conformational change in the spatial relationship between mIg heavy chain and Igβ is dependent on the cytoplasmic tail.
DOI: https://doi.org/10.7554/eLife.42271.017

we combined site-specific labeling methodology and FRET-based assay to address above-mentioned questions. By inserted short peptides that allow targeted incorporation of fluorophores into different regions, we achieved site-specific labeling within BCR complex to enable monitoring distance change between two labeled sites through FRET. We checked two FRET pairs: one pair represented conformational change within mIg heavy chain, whereas the other pair demonstrated conformational change in the spatial relationship between mIg and Igα/β. We also compared the conformational change between antigen-bound IgM- and IgG-BCR.

We firstly found decreased FRET efficiency between N terminus and Cμ2 domain within mIg of IgM-BCR upon antigen engagement. Notably, decreased proximity between this two sites were discovered in both 293T cells and A20II1.6 B cells, indicating the independence of the conformational change on downstream signaling after antigen stimulation. Since these two regions locate across the hinge region linking Fab and Fc fragment of mIg, distance change between N terminus and Cμ2

domain observed in this study might involve stretching or bending of the hinge region. Interestingly, the level of conformational change (as indicated by the FRET efficiency) within mIg of IgM-BCR was positively associated with the strength of B cell activation. However, such conformational change was not observed in IgG-BCR. We also found that conformational change within mIg was dependent on the absence of IgG cytoplasmic tail, while a reproducible trend was consistently observed that the presence of IgG cytoplasmic tail slightly, but non-significantly increased the basal distance between N terminus and Cμ2 (or Cγ2) domain of VRC01-BCR without antigen binding. Combining all these findings, we speculated that in the absence of IgG cytoplasmic tail, antigen engagement induced conformational change in the spatial relationship between N terminus and Cμ2 domain within mIg of IgM-BCR, while greater conformational change was associated with higher activation level in B cell. Potentially, conformational change in the spatial relationship between N terminus and Cμ2 domain within mIg of antigen-bound IgM-BCR was then transmitted to membrane-proximal domain, while this transduction was achieved in IgG-BCR without change of spatial distance between N terminus and Cγ2 domain. Extracellular antigen signal transmitted to membrane-proximal region might trigger local conformational change in Fc domain of mIg, inducing an orientation to expose a clustering interface on the Cμ4 (or Cγ3) domain (*Tolar et al., 2009*) (*Figure 6*).

In our experimental system, intermolecular FRET occurred between neighboring BCR complexes as BCR cluster formation might interfere the FRET signals between interested areas within one BCR complex, although according to the research in TCR, intermolecular FRET was very low (*Lee et al., 2015*). To solve this, when exploring the conformational change within BCR engaged with membrane antigen, we analyzed data from cells with equivalent mean density of donor and acceptor (indicated by mFI). We also deconvolved the intramolecular FRET and intermolecular FRET by expressing untagged BCRs to dilute tagged BCRs in cells, proving the FRET change mainly resulted from intramolecular FRET, rather than intermolecular FRET.

Antigen binding leads to phosphorylation of ITAMs in Igα/β heterodimer, which is noncovalently linked with mIg molecule. In the proposed conformational change model of BCR-mediated B cell activation, signals generated from extracellular stimulation will finally reach at Igα/β through mIg component. We discovered decreased FRET efficiency between Fc region of mIg and Igβ upon antigen engagement in IgG-BCR expressing 293T cells and A20II1.6 B cells. Similarly, conformational change in the spatial relationship between these two sites was also dependent on mIg cytoplasmic tail (*Figure 6*). Interestingly, we found that in VRC01-BCR without antigen binding, the presence of IgG cytoplasmic tail increased the basal proximity between mIg and Igβ, indicating IgG cytoplasmic tail might affect the interaction between mIg and Igβ in the absence of antigen. This effect of IgG cytoplasmic tail might involve its interaction with the inner leaflet of the plasma membrane (*Chen et al., 2015*; *Wan et al., 2018*). Furthermore, the extent of conformational change in the spatial relationship between mIg and Igβ was positively associated with the B cell activation level. Increased spatial distance between mIg and Igα/β of IgG-BCR might generated from the dissociations of IgG cytoplasmic tail and Igα from the plasma membrane (*Lee and Tolar, 2013*; *Chen et al., 2015*) (*Figure 6B*). Meanwhile, it has been reported that the extracellular domains of Igα/β preferentially recognize the Fc region of IgM compared with IgG (*Radaev et al., 2010*), potentially inhibit the dissociation between mIg and Igβ in IgM-BCR. In line with previous findings, reduced proximity between extracellular domains of mIg and Igβ might induce the increased distance between cytoplasmic domains of mIg and Igα/β heterodimer, facilitating the formation of 'umbrella opening form' in cytoplasmic domains of antigen-activated BCR (*Tolar et al., 2005*).

One limitation of our study is that the physiological meanings of the observed conformational changes within extracellular domain of BCR remain unclear. Although the association between conformational changes of BCR and the strength of B cell activation was found, it is still premature to conclude that the antigen-induced conformational changes of BCR are causal for triggering downstream signaling of B cell. The actual role (activating or inhibitory effect) of the reported BCR conformational changes in B cell activation can be complex depending on the context of the antigens, the concept of which can be reflected from the investigations on conformational changes of other membrane receptors, such as G-protein-coupled receptor (GPCR). Agonist and antagonist will induce distinct conformational change of GPCR, indicating different conformational changes within GPCR might trigger different physiological functions (*Vafabakhsh et al., 2015*; *Olofsson et al., 2014*). Specifically, the observation that conformational change between N terminus and Cμ2 domain within mIg of IgM-BCR (which was positively associated with the B cell activation level) did not occur in the

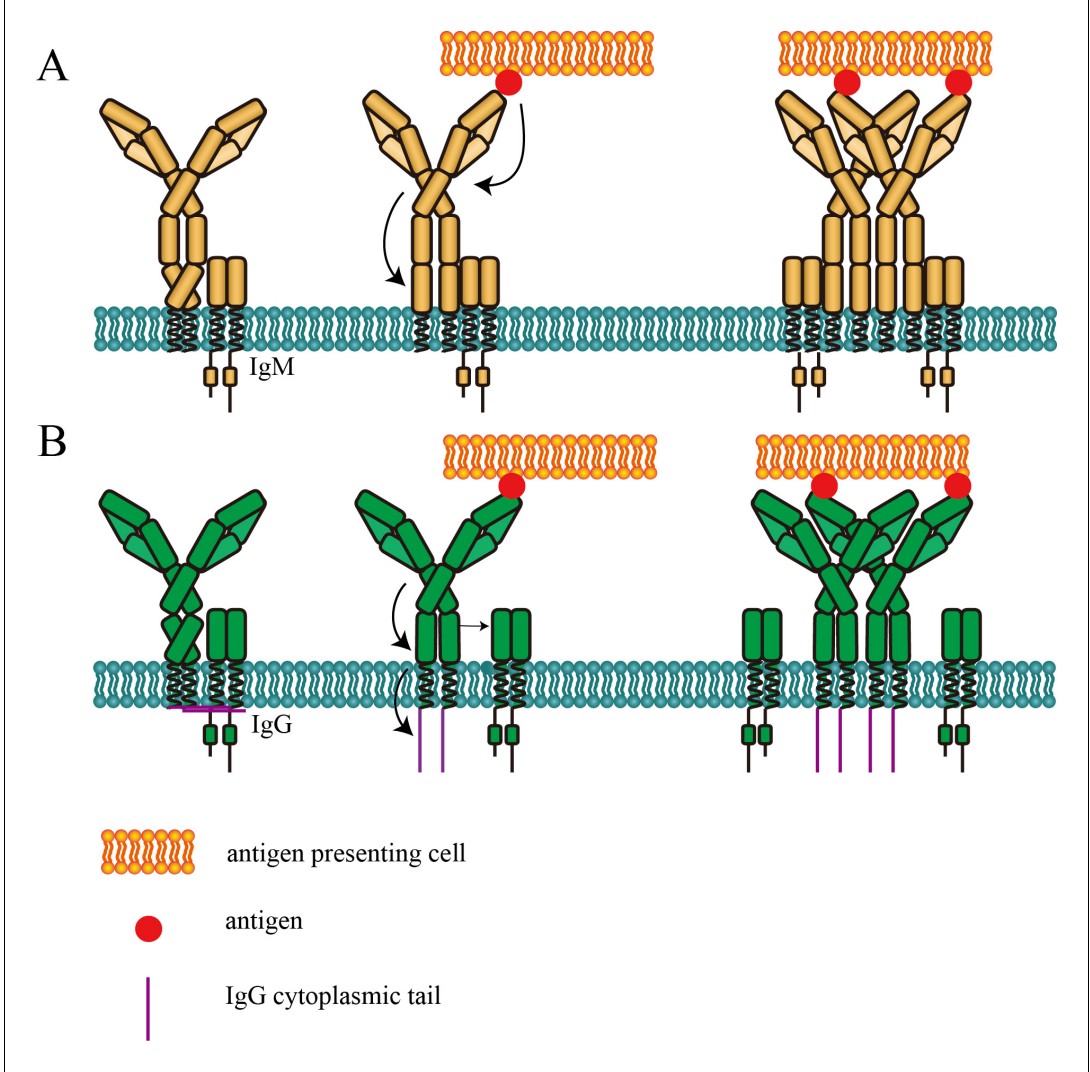

**Figure 6.** Cartoon models of BCR conformational change. (**A**) Conformational change model of IgM-BCR activation. Upon antigen engagement, distance between N-terminus and Cμ2 domain within mIgM increases, which might be resulted from stretching or bending of the hinge region. This conformational change might trigger the potential conformational change within Fc domain to expose the clustering interface in Cμ4 domain. (**B**) Conformational change model of IgG-BCR activation. After binding with antigen, potential conformational change within Cγ3 domain occurs independently of the distance change between N-terminus and Cγ2 within mIgG. Then the proximity between mIg and Igα/β decreases, which might be involved with the dissociation between IgG cytoplasmic tail plasma membrane. The exposure of the clustering interface in Cμ4 (or Cγ3) domain will then promote BCR clustering.

DOI: https://doi.org/10.7554/eLife.42271.018

presence of IgG cytoplasmic tail seems to contradict with the previous findings, which report the enhancement of IgG cytoplasmic tail on the antigen-induced B cell activation (*Chen et al., 2015*; *Liu et al., 2010b*; *Wakabayashi et al., 2002*). One possible explanation is that conformational change in the spatial relationship between N terminus and Cμ2 (or Cγ2) domain is only required for IgM-BCR (or IgG-BCR equipped with IgM cytoplasmic tail) to transmit antigen-binding signal to membrane-proximal domain, inducing an orientation to expose a clustering interface on the Cμ4 (or Cγ3) domain (*Tolar et al., 2009*). In contrast, this conformational change is probably not required for IgG-BCR (or IgM-BCR equipped with IgG cytoplasmic tail) to transmit the extracellular signal of antigen-binding to induce the potential conformational change near the Cγ3 (or Cμ4) domain. This hypothesis seems to be consistent with the previous findings that IgG cytoplasmic tail lowers the threshold of mechanical force to induce IgG-BCR activation (*Wan et al., 2015*; *Wan et al., 2018*). Considering that the mechanical force might be closely related with the conformational change, it is

reasonable to speculate that IgG cytoplasmic tail might lower the threshold of B cell activation through bypassing the required conformational change between N terminus and Cμ2 domain within mIg in IgM-BCR. Thus, it will be constructive to investigate the association between mechanical force and conformational change of BCR in the future study. Another possible explanation is that spatial distance change between N terminus and Cμ2 domain of mIg in IgM-BCR could be interpreted as an inhibitory mechanism. Antigen-induced conformational change between N terminus and Cμ2 domain might inhibit further BCR activation through an unexplored mechanism. An interesting hypothesis is that distance change between N terminus and Cμ2 domain might affect the BCR-antigen bond lifetime. As accumulation of bond lifetimes of TCR-pMHC is required for T cell signaling according to a published study (*Liu et al., 2014*), it shall be intriguing to explore the effect of antigen-induced conformational change on BCR-antigen bond lifetime.

The mechanism by which IgG cytoplasmic tail influences the antigen-induced conformational changes of BCR is also worth further exploration. We speculate that through its interaction with cell membrane (*Chen et al., 2015*), IgG cytoplasmic tail might alter the basal conformational state of BCR complex (in fact, we found that it can significantly affect the distance between mIg and Igα/β in the absence of antigen), or alter the mechanical force sensing capability of BCR (*Wan et al., 2018*). Analyzing the structure of BCR complex by cryo-electron microscopy might uncover the exact mechanism.

In summary, this study reports antigen-induced conformational changes within extracellular domain of BCR, providing evidence for the 'conformational change model', which has been proposed as a possible mechanism to explain the initiation of antigen-induced B cell activation. Furthermore, these conformational changes are associated with BCR-mediated B cell signaling. These observations may offer a perspective for the further investigation of triggering and regulating mechanisms in BCR activation and B cell function. In addition, our study provides a framework for exploring ligand-induced intramolecular or intermolecular events in BCR and other immune receptors.

## Materials and methods

### Plasmids, cells, antibodies and reagents

Plasmids carrying components of BCRs were all constructed in pHAGE backbone. In plasmids carrying tagged mIg heavy chains, ybbR tag (DSLEFIASKLA) and/or tetracysteine tag (FLNCCPGCCMEP) were inserted. In plasmid carrying tagged Igβ, tetracysteine tag was inserted. Insertion sites for the ybbR tag or tetracysteine tag are indicated in bold in the following table.

| Construct | Amino acid sequence |
| --- | --- |
| ybbR tag in N terminus of VRC01-BCR | **DSLEFIASKLA**QVQLVQSGGQ |
| tetracysteine tag in Cμ2 of VRC01-IgM-BCR | VPPRDGFFGN**FLNCCPGCCMEP**PRKSKLICQA |
| tetracysteine tag in Cγ2 of VRC01-IgG-BCR | KTHTCPPCPA**FLNCCPGCCMEP**PELLGGPSVF |
| tetracysteine tag in Igβ | IYFCQQKCNN**FLNCCPGCCMEP**TSEVYQGCGT |

Human IgM and IgG constant regions of heavy chains were fused with VRC01-specific variable region to construct mIg heavy chains of VRC01-IgM- and IgG-BCR, respectively. Plasmids carrying mutant Ig cytoplasmic tails of heavy chains were constructed based on the WT versions. Human Igκ constant region was fused with VRC01-specific variable region to construct mIg light chain of VRC01-BCR. A20II1.6 B cells or 293T cells were co-transfected with mIg heavy chain, mIg light chain, human Igα and human Igβ by electroporation (Lonza) or calcium phosphate transfection, respectively. A20II1.6 B cell line (*Liu et al., 2010c*) and Ramos B cell line (RRID:CVCL_0597) were gifts from Dr. Susan K. Pierce (NIAID, NIH, USA) that were originally purchased from ATCC (USA) and maintained in complete RPMI-1640 medium with 10% heat-inactivated fetal bovine serum (FBS), penicillin and streptomycin antibiotics (Invitrogen). 293T cells were purchased from Cell bank (Chinese academy of sciences, Shanghai) and cultured in DMEM containing 10% FBS, penicillin, and streptomycin antibiotics. V51 monomer and V51 trimer antigens were gifts from Dr. Zhang Linqi (Tsinghua University, Beijing, China). All cell lines used in this study were negative for mycoplasma contamination test based on PCR detection method. Goat non-specific IgG, Alexa Fluor 647 conjugated Fab anti-

human IgM Fc5μ and Alexa Fluor 647 conjugated Fab anti-human IgG Fcγ were purchased from Jackson ImmunoResearch. CoA 488 and Sfp phosphopantetheinyl transferase (SFP synthase) used for ybbR tag labeling were purchased from NEB. ReAsH-EDT2 for tetracysteine tag labeling was purchased from Santa Cruz. Chemical reagents used in all experiments were purchased from Sigma Aldrich.

## Purification of soluble VRC01-IgM

WT or tagged VRC01-IgM monomers (without J chain) fused with a His$_6$ tag at C terminus were expressed in 293 F cells. The culture supernatants were purified by Ni$^{2+}$ column and were further purified by Superdex 200 column (GE Healthcare).

## Site-specific labeling of BCR and soluble Ig

Labeling of ybbR tag in cells was achieved according to published protocols (*Yin et al., 2006*). Briefly, 293T or A20II1.6 B cells expressing ybbR-tagged BCR were incubated with 1 μM SFP synthase, 10 mM MgCl$_2$, 2 μM CoA 488 in HBSS buffer for 20 min at room temperature. For tetracysteine-tagged BCR labeling, cells were pre-incubated with 5 mM 2-mercaptoethanesulfonate (MES) and 0.5 mM tris(carboxyethyl)phosphine (TCEP) at 37°C for 20 min, then the pre-treated cells were stained in HBSS buffer containing 1 μM ReAsH-EDT2 and 25 μM 2,3-dimercapto-1-propanol (BAL) for 10 min at 4°C, then washed by 100 μM BAL (*Hoffmann et al., 2010*). For Fc5μ or Fcγ fragment labeling, cells were stained for 5 min at 4°C with Alexa Fluor 647 conjugated Fab anti-human IgM Fc5μ or Alexa Fluor 647 conjugated Fab anti-human IgG Fcγ, respectively. For ybbR tag labeling in soluble tagged soluble VRC01-IgM, purified proteins were incubated with 1 μM SFP synthase, 10 mM MgCl$_2$, 5 μM CoA 488, 10 mM HEPES for 30 min at 37°C. For ReAsH labeling, tagged soluble VRC01-IgM molecules were treated with 1 μM ReAsH-EDT2 in HBSS buffer for 30 min at room temperature. Labeled proteins were desalted using Zeba Spin Desalting Columns (Thermo Fisher).

## Ca$^{2+}$ mobilization analysis

To detect the intracellular Ca$^{2+}$ mobilization of A20I1.6 B cells expressing tagged VRC01-BCR, cells were processed with 1 μM Fluo4-AM (Invitrogen) and 2.5 mM probenecid (Invitrogen) at 30 °C for 30 min in HBSS (with Ca$^{2+}$ and Mg$^{2+}$) buffer. Then cells were incubated at 37 °C for 15 min in HBSS buffer. Treated cells were stimulated with 100 nM HIV-1 gp120 antigen, calcium flux was measured by BD Accuri C6 cytometer and analyzed by FlowJo software.

## V51 monomer binding measurement

293T cells expressing WT or tagged VRC01-BCR were firstly incubated with 50 nM V51 monomer (fused with poly-histidine tag) for 10 min at room temperature followed by washing with PBS. Then cells were treated with anti-His tag antibody (mouse IgG1, Biodragon Immunotechnologies, Beijing) at room temperature for 10 min. After washed with PBS, cells were stained with Alexa Fluor 488-conjugated-affinipure goat anti-mouse IgG, Fcγ fragment specific (minimal cross-reaction to human, bovine, and horse serum proteins, Jackson ImmunoResearch) at 4°C for 10 min. Results were acquired by BD Accuri C6 cytometer and analyzed by FlowJo software.

## Preparation of antigen-containing coverslip

Glass coverslips (VWR International) were pretreated with acid (H$_2$SO$_4$:H$_2$O$_2$ = 7:3) and then were washed and dried before glued to the 8-well chamber frame (Nunc Lab-Tek chamber). For experiments in which cells were stimulated with membrane V51 monomer antigen, 0.1 mM Ni$^{2+}$-containing planar lipid bilayer (PLB) were prepared as our published study (*Zhang et al., 2013*) and loaded on coverslip for 30 min at room temperature followed by washing with PBS, then 20 nM V51 monomer (fused with poly-histidine tag) were added and incubated for 30 min at room temperature. After addition of antigen, chambered coverslip was washed with PBS and then was blocked with 1% casein in PBS for 30 min at 37°C. Finally, the labeled cells were loaded on the antigen surface for reaction at 37°C for 3 min and fixed with 4% paraformaldehyde (PFA). In control groups without antigen, coverslip was pre-treated with acid, after which Ni$^{2+}$-PLB were added and incubated, then coverslip was blocked by 1% casein. Finally, the cells were loaded on the surface without V51 molecules.

For experiments in which soluble VRC01-IgM molecules were stimulated with membrane V51 monomer antigen, coverslip was similarly pre-treated with acid, incubated with $Ni^{2+}$-PLB followed by V51 molecules, and blocked with 1% casein. Then the soluble VRC01-IgM molecules were added on the antigen surface. After reaction for 10 min at 37°C, soluble VRC01-IgM molecules were imaged by confocal fluorescence microscopy. In control groups without antigen, coverslip was pre-treated with acid and blocked with 1% casein. Then soluble VRC01-IgM molecules were loaded on coverslip without V51 molecules and were incubated to allow the non-specific adherence on the surface before imaging.

## Molecular imaging by TIRFM and confocal fluorescence microscopy

TIRFM images were captured by an Olympus IX-81 microscope equipped with a TIRF port, an Olympus 100 × 1.49 NA oil objective lens, and an Andor iXon +DU-897D electron-multiplying charge-coupled device camera. Image acquisition was controlled by MetaMorph software (Molecular Devices) with a 100 ms exposure time for a 512 × 512 pixel image. Confocal images were acquired by a FLUOVIEW FV1000 confocal laser scanning microscope (Olympus) with a 60 × 1.42 NA oil objective lens. All the images were acquired while keeping the cells in HBSS and were analyzed with ImageJ (National Institutes of Health). The mFI of donor fluorophores, acceptor fluorophores and signaling molecules were calculated as described previously (*Liu et al., 2010a*; *Liu et al., 2010b*; *Liu et al., 2010c*). When acquiring images of resting cells without antigen binding, we selected cells with high fluorescent intensity to match resting cells to antigen-stimulated cells.

## FRET measurement by acceptor photobleaching method

Images of stimulated and fixed cells on antigen-containing coverslip were acquired by TIRFM, while images of soluble Ig molecules were obtained by confocal fluorescence microscopy. FRET pairs used here included CoA 488 (donor) +ReAsH (acceptor), and ReAsH (donor) +Alexa Fluor 647 (acceptor). All the images were processed with ImageJ. FRET efficiency was calculated with the formula: FRET efficiency = $(DQ - Q)/DQ$, where DQ and Q representing dequenched and quenched donor fluorescence intensity, respectively. To match the fluorescent intensity of donor and acceptor in resting cells and antigen-stimulated cells for analysis, resting cells with relatively low fluorescent intensity and antigen-stimulated cells with relatively high fluorescent intensity were excluded.

## FRET measurement by FLIM

Time-correlated single-photon counting (TCSPC) system (PicoQuant) was used for FLIM-FRET measurement following the published protocol (*Sun et al., 2011*; *Jahn et al., 2015*). Dually labeled (CoA 488 + ReAsH, ReAsH +Alexa Fluor 647) and donor only labeled (CoA 488, ReAsH) cells, or dually labeled (CoA 488 + ReAsH) and donor only labeled (CoA 488) soluble VRC01-IgM molecules in HBSS buffer were loaded to antigen-containing coverslip. FLUOVIEW FV1200 confocal fluorescence microscope consisting of an inverted microscope (IX 83, Olympus) equipped with an Olympus UPLSAPO 60 × NA 1.3 oil immersion objective was used to collect the fluorescence signal. The donor fluorophore (CoA 488 or ReAsH) was excited with 480 nm or 560 nm laser and the donor signal was detected with 520/35 nm or 615/60 nm bandpass filter, respectively. In all experiments, the laser power was adjusted to achieve average photon counting rates $\leq 10^5$ photons/s. All FLIM data acquisition and analysis were performed with SymPhoTime 64 software.

## Intracellular immunofluorescence staining and molecular imaging

The recruitment of signaling molecules into the immunological synapse of B cells stimulated by membrane antigen was imaged by TIRFM by following our previously published protocol (*Liu et al., 2010a*; *Liu et al., 2010b*; *Liu et al., 2010c*). In brief, stimulated A20II1.6 B cells fixed on antigen-containing chambered coverslip were permeabilized with 0.1% Triton X-100 and then blocked with 0.1 mg/ml goat non-specific IgG. Subsequently, cells were stained with phosphor-CD79A (Tyr182) primary antibody (Cell Signaling Technology) at room temperature for 1 hr. After washing with PBS, cells were stained with secondary antibody Alexa Fluor 488 or 647-conjugated F(ab')$_2$ goat anti-rabbit IgG H + L (Invitrogen) at room temperature for 45 min. After washing with PBS, the cells were imaged by TIRFM and analyzed by ImageJ.

## Acknowledgements

We acknowledge the assistance of Image Core Facility, Technology Center for Protein Sciences, Tsinghua University, for assistance of using FV1200 LSCM with Picoquant FLIM/FCS system. This work is supported by funds from National Science Foundation China (81825010, 81730043, 81621002 and 31811540397).

## Additional information

### Funding

| Funder | Grant reference number | Author |
| --- | --- | --- |
| National Natural Science Foundation of China | 81825010 | Wanli Liu |
| National Natural Science Foundation of China | 81730043 | Wanli Liu |
| National Natural Science Foundation of China | 81621002 | Wanli Liu |
| National Natural Science Foundation of China | 31811540397 | Wanli Liu |

The funders had no role in study design, data collection and interpretation, or the decision to submit the work for publication.

### Author contributions

Zhixun Shen, Conceptualization, Data curation, Validation, Investigation, Methodology, Writing—original draft, Writing—review and editing; Sichen Liu, Investigation, Methodology, Writing—review and editing, Assisted with all experiments; Xinxin Li, Investigation, Methodology, Writing—review and editing, Carried out protein expression and purification experiments; Zhengpeng Wan, Conceptualization, Writing—review and editing; Youxiang Mao, Investigation, Methodology, Assisted with protein labeling experiments; Chunlai Chen, Writing—review and editing; Wanli Liu, Conceptualization, Resources, Supervision, Funding acquisition, Validation, Methodology, Project administration, Writing—review and editing

### Author ORCIDs

Wanli Liu (iD) https://orcid.org/0000-0003-0395-2800

### Decision letter and Author response

Decision letter https://doi.org/10.7554/eLife.42271.021
Author response https://doi.org/10.7554/eLife.42271.022

## Additional files

### Supplementary files

• Transparent reporting form
DOI: https://doi.org/10.7554/eLife.42271.019

### Data availability

All data generated or analysed during this study are included in the manuscript and supporting files.

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
