## [Decision Letter]

Thank you for submitting your article "Conformational change of B cell receptor in the initiation of B cell activation" for consideration by *eLife*. Your article has been reviewed by Arup Chakraborty as the Senior Editor, a Reviewing Editor, and three reviewers. The following individuals involved in review of your submission have agreed to reveal their identity: Pavel Tolar (Reviewer #1); Balbino Alarcon (Reviewer #2).

The reviewers have discussed the reviews with one another and the Reviewing Editor has drafted this decision to help you prepare a revised submission.

The authors generate some intriguing data that suggest conformational changes of BCR that are induced by antigen on the cell surface, but there is some concern that the authors try to take on too many different questions related to this without establishing a strong enough foundation to analyze the FRET signatures. If the authors were to focus on the site-specific labeling system set up in VRC01 IgM and fully characterize it with all necessary controls, they could have a strong study that may provide an unprecedented picture IgM extracellular domain conformation change on binding of a monovalent antigen.

Essential revisions:

1) All reviewers were concerned about the use the polyclonal Fab in FRET studies of IgM and IgG. There are 6 Ig domains in IgM Fc and 4 in IgG Fc to which these reagents could bind. So, these reagents were considered inappropriate for the task in terms of both steric effects of Fab multiple binding and the uncertainty as to the location of binding if all sites were not saturated. Unfortunately, none of the experiments using these reagents will be appropriate for publication at the level *eLife* aspires to. On the other hand, the site specific labeling strategy in the VRCO1 BCR were considered sufficiently site specific and well characterized functionally to be of great use. The authors are urged to build a better foundation around VRC01 binding to gp120 monomers (verify). The reviewers have made some suggestions below regarding possible ways forward that might be achievable in a revised manuscript in 2 months, however, the authors are given latitude to take other approaches that provide interpretable FRET data from a well defined system to provide convincing evidence of conformational changes.

2) The authors should be able to generate CoA488 and ReAsH labeled soluble IgM and study the FRET in solution and the effect of antigen binding with absolute quantification. While this is an IgM antibody, there are ways to produce these in 293 cells without or with J-chain to generate monomers or pentamers. This would enable a systematic study of the conformational dynamics of the receptors in solution with and without ligand and binding and forced clustering. The system could be used to calibrate the FRET signature of cell surface FRET. These experiments could be done in bulk, single cell or microscopy modes.

3) The authors should verify that gp120 is monomeric and could determine the consequences of gp120 binding to the monomeric or pentameric soluble IgM in terms of any evidence for generating of higher order structures, which would suggest that the gp120 is not monomeric or can be induced to form oligomers when engaged by the Fab. But the resulting multimers could be characterized by gel filtration or analytical ultracentrifugation.

4) Parts of Figure 1, Figure 2Figure 6 and Figure 7 could be used as a core for a revised paper that provides a better-controlled study that focuses on the gp120 system. The authors may have sufficient time to generate results with solid phase gp120, which can be attached to a solid substrate or presented in supported lipid bilayers.

The revisions should also address:

1) Does the BCR undergo a conformational change detected in the ectodomains upon antigen binding? With soluble IgM they might hope to be able to model the FRET and even estimate average distances.

2) Does the conformational change correlate with downstream activation signaling pathways? The experiments with IgGa/b labeling in the gp120 system may shed light on this.

3) The site specific labeling of the BCR is a novel and useful tool. But to interpret the donor-dequenching data, it is important to know the labelling efficiency with the donor and acceptor. The relative stoichiometry of the labels is particularly critical for comparing the different conditions and BCR constructs. FLIM is a good way to determine the level of FRET irrespectively of labelling stoichiometry, and the authors do use it in one experiment, but unfortunately, they do not derive the information about the fraction of donor molecules in FRET and their FRET level. This could be achieved with soluble IgM and lessons applied to understanding what is happening on the cell surface.

4) Controls for the FRET photobleaching measurements using donor only and acceptor only samples were missing. These are important as there is often some bleedthrough and also some photobleaching-induced photoconversion. Given that the antigen-induced FRET changes are small, these additional corrections may be significant.

5) Even if the authors don't perform the calibration studies with soluble IgM, they need to deconvolve intermolecular and intramolecular FRET. Its felt that soluble monomers and pentamers will provide critical insight into this, but otherwise another method to look separate out these effects is needed.

6) The authors should provide a better structural model for the conformational changes implied by the FRET measurements. It seems that there are limited number of ways to explain the results and authors should be able to provide schematics to systematically explore these, which may clarify experiments needed to disambiguate these in this study or the future.

[Editors' note: further revisions were requested after re-review, as described below.]

Thank you for resubmitting your work entitled "Conformational change within the extracellular domain of B cell receptor in B cell activation upon antigen binding" for further consideration at *eLife*. Your revised article has been favorably evaluated by Arup Chakraborty (Senior Editor), a Reviewing Editor, and three reviewers.

The manuscript has been improved but there are some remaining issues that need to be addressed before acceptance, as outlined below:

Summary:

There is a general agreement that the FRET measurements have been improved by the revision and that conformational changes in the extracellular domain are being detected. However, all reviewers felt that the interpretation that these conformational changes are causal for BCR triggering is not demonstrated. In fact, one possibility that was raised is that the conformational change is only observed in the receptor that lacks the cytoplasmic tail of IgG, which is known to have signaling activity. Thus, one possibility not explored by you is that the conformational change results in some mode of negative feedback that is averted by signaling through the cytoplasmic tail of IgG isotype, but not IgM.

Essential revisions:

1) Doubts remain on the physiological meanings of the observed effects, especially with regard to the initiation of BCR signaling. The main conclusion of the authors is that intramolecular conformational changes in surface Ig molecules transport the information of extracellular antigen binding across the plasma membrane to induce intracellular BCR signaling (see subsection “Conformational changes were distinct in soluble IgM and IgM-BCR” and Discussion section). This main conclusion is not justified by the data presented. FRET experiments shown in Figure 3 indicate that those BCR classes with enhanced signaling potency (i.e. those with an Ig-γ tail) did NOT show any intramolecular conformational changes. Hence, if the observed conformational changes are a physiologically meaningful mechanism, it is not to "promote" or "facilitate the intracellular signaling in B cells" as suggested by the authors. In fact, it is well established that surface IgG has a greatly improved signaling capacity compared to IgM because the latter lacks a cytoplasmic signaling tail (Martin and Goodnow, 2002; Wakabayashi et al., 2002; Engels et al., 2009; Liu et al., 2010). These reports are not discussed in the present manuscript. If anything, the presented results could be interpreted as an inhibitory mechanism since the changes are specific to BCRs with reduced signaling capability (i.e. BCRs without an Ig-γ tail). Also, the data shown in Figure 2J are not really supporting the idea that conformational changes in the extracellular domains of IgM drive intracellular signaling, since it is not clear how the 'immunological synapses' of cells with high and low FRET signals look like. A reduced FRET signal could simply be an indicator for efficient antigen binding and efficient BCR clustering, which would explain why those cells have enhanced amounts of phospho-CD79.

Another limitation of the study that remains is the lack of a mechanistic explanation to the main observation, which is that both intra- and intermolecular conformational changes in either surface IgM or the IgG-BCR complex depend on the absence or presence of an intracellular Ig-γ tail. How can the transfer of the IgG tail to IgM prevent a conformational change between Fab and Cμ2 domain? An inside out signaling mechanism can be excluded, since this phenomenon is also observed in 293 cells. The illustrations shown in Figure 7 and the Discussion section do not provide any clues and in combination with the unclear situation regarding the physiological meaning of the observed effects, the impact of the entire study on the field remains limited.

2) The interpretation of the changes in FRET between the N-terminus of the Ig and the Cm2 (or Cg2) is mostly with respect to the putative changes induced by antigen that propagate to the intracellular domains. However, this FRET measures distances across the Fc-Fab hinge region, so it is likely that the majority of the movement here involves some stretching or bending at the hinge as two antigens are engaged by the two Fab arms. This should be at least mentioned and included as one interpretation in the manuscript. It is quite possible to imagine that the average location of the Fab arms in IgM vs IgG is different and therefore a different effect may be observed after antigen binding. The effect of the cytosolic tail is potentially very interesting as it suggests that its presence prevents the antigen-induced change. However, in Figure 3, the data show that it is the basal levels of FRET that are changing between constructs, not the antigen-stimulated. Thus, there may be some effect of the intracellular tail on the orientation of the Fab fragments without antigen binding, not necessarily on "conformational changes" induced by antigen. Similar effects of the tail can be seen in FRET with the Igbeta in Figure 6. This again should be included in the authors' interpretation of the data. I think it is a bit premature to conclude that these conformational changes are causal in triggering signaling.

[Editors' note: further revisions were requested prior to acceptance, as described below.]

Thank you for resubmitting your work entitled "Conformational change within the extracellular domain of B cell receptor in B cell activation upon antigen binding" for further consideration at *eLife*. Your revised article has been favorably evaluated by Arup Chakraborty (Senior Editor), a Reviewing Editor, and three reviewers.

The manuscript has been improved but there are some remaining issues that need to be addressed before acceptance, as outlined below:

There is a general feeling that you have shown a link between the intracellular composition of the receptor and conformation of the extracellular antigen binding component of the receptor. The mechanism by which this is happening is not clear. We would like to publish your manuscript, but wanted to provide an opportunity for you address some specific comments to present the findings with maximum clarity in light of well known tissue with assessing statistical significance based on p values and related issue of how cells are selected for analysis.

Essential revisions:

1) The major issue is still with the two parts of the paper that asses the effect of the intracellular IgG tail on the FRET in the extracellular domain. Since there is little mechanistic understanding how this can happen, it is important to report on the observations carefully. In Figure 5, based on t-tests, the authors conclude that adding the IgG tail to IgM or removing the tail from IgG changes the FRET levels between the Fc and Ig-β in resting cells. In contrast, in Figure 3, the authors conclude that the effect of the tail on the resting FRET between the Fc and the Fab is not significant and therefore nonexistent. This, however, seem to be just a matter of a p-value cutoff for this particular test, as there is a trend for the FRET changing with p as low as 0.063 in one of the conditions. This creates an odd conclusion – the IgG tail changes neither the resting nor the antigen-induced FRET, yet it changes whether the two are statistically separable from each other. First, it needs to be disclosed if the data in the supplementary figures (Figure 3—figure supplement 1 C,D, or Figure 5—figure supplement 1C,D) are from the same experiments as in the main figures (Figure 3, Figure 5). If yes, the statistical test are not adequate. The whole experiment should be analyzed with ANOVA with corrections for multiple hypothesis testing. Even if the statistical significance is not illuminating, I think it would be good to conclude whether there is a reproducible trend for a change in the FRET levels of the individual conditions or whether simply the data are not strong enough to conclude as to why the tail changes the "response" to antigen.

2) The authors are careful in matching the fluorescent intensity of the cells analysed in the FRET experiments and they do this by selecting cells for the analysis. Presumably there is an increase in BCR fluorescence in cells interacting with antigen. It would be worth explaining in the methods exactly how the cells were selected – are antigen stimulated cells matched to resting or vice versa?

---

## [Author Response]

The authors generate some intriguing data that suggest conformational changes of BCR that are induced by antigen on the cell surface, but there is some concern that the authors try to take on too many different questions related to this without establishing a strong enough foundation to analyze the FRET signatures. If the authors were to focus on the site-specific labeling system set up in VRC01 IgM and fully characterize it with all necessary controls, they could have a strong study that may provide an unprecedented picture IgM extracellular domain conformation change on binding of a monovalent antigen.

We appreciate the comments from the editors and the reviewers to our study. As described in detail below, we have tried to address all the questions and comments from the reviewers. Our responses to each individual comment have been integrated into the revised manuscript by adding new data and/or editing the manuscript. We hope our revised manuscript will fulfill the requirements needed for publication in *eLife*.

Essential revisions:1) All reviewers were concerned about the use the polyclonal Fab in FRET studies of IgM and IgG. There are 6 Ig domains in IgM Fc and 4 in IgG Fc to which these reagents could bind. So, these reagents were considered inappropriate for the task in terms of both steric effects of Fab multiple binding and the uncertainty as to the location of binding if all sites were not saturated. Unfortunately, none of the experiments using these reagents will be appropriate for publication at the level eLife aspires to. On the other hand, the site specific labeling strategy in the VRCO1 BCR were considered sufficiently site specific and well characterized functionally to be of great use. The authors are urged to build a better foundation around VRC01 binding to gp120 monomers (verify). The reviewers have made some suggestions below regarding possible ways forward that might be achievable in a revised manuscript in 2 months, however, the authors are given latitude to take other approaches that provide interpretable FRET data from a well defined system to provide convincing evidence of conformational changes.

Thank you for all of these insightful suggestions. We agree that for determination of intramolecular FRET between N terminus and Fc region within mIg heavy chain of BCR, site-specific labeling of both positions is necessary. Thus, for measurement of conformational change in the ectodomains of IgM-BCR, we took the suggestion and focused on the VRC01-IgM-BCR with ybbR tag inserted in N terminus and tetracysteine tag inserted in Cμ2 region (Figure 1A). In addition, to compare the conformational change of IgM- and IgG-BCR, we constructed VRC01-IgG-BCR with ybbR inserted in N terminus and tetracysteine tag inserted in Cγ2 region (Figure 1—figure supplement 1) in the revised manuscript. Based on these two VRC01-BCR constructs, we intensely examined the acquired FRET signal of BCR upon binding with HIV-1 gp120 derived-V51 monomer presented on the surface of planar lipid bilayer (Please see Essential revision 3 and 4). Importantly, conformational changes within site-specifically labeled ectodomains of different BCR constructs observed in this revised manuscript are largely consistent with observations generated with polyclonal Fab in NP antigen system in the original manuscript (Figure 2, Figure 3).

For measurement of FRET between Igα/β and mIg heavy chain, we tried to site-specifically label Igβ and mIg heavy chain in IgM-BCR respectively using two strategies: tetracysteine tag for Igβ labeling and ybbR tag for Fc region of mIg labeling, or tetracysteine tag for Fc region of mIg labeling and ybbR tag for Igβ labeling. However, after trying over 10 sites in Fc region of mIg (6 sites in Cμ2, 5 sites in Cμ3) and 4 sites in Igα/β (2 sites in Igα, 2 sites in Igβ), we were still unable to find any positions with ybbR tag insertion can be efficiently labeled by CoA 488 fluorophore (please see representative images Author response image 1). According to the structural study of Igα/β and its assembly with mIg (Radaev et al., 2010), the interaction between Igα/β and Cμ2-Cμ4 of mIg in IgM-BCR might hamper the label of ybbR tag. Considering that FRET between Igβ and mIg in BCR complex results from intermolecular FRET, it is reasonable to use polyclonal Fab fragment of antibody to obtain average FRET (please see Author response image 2) between two distinct neighboring molecules, as several published papers used fluorophore-conjugated polyclonal Fab to measure the intermolecular FRET efficiency, including the intermolecular FRET between neighboring BCR molecules (Tolaret al., 2005; Verveer et al., 2000; Veeriah et al., 2014; Liu et al., 2010). Thus, we used polyclonal Fab for Fc region staining to measure FRET between Igβ and mIg heavy chain in VRC01-BCR, generating data that are consistent with those observations generated with polyclonal Fab in NP antigen system in the original manuscript (Figure 5, Figure 6).

**Author response image 1. respfig1:** Labeling of ybbR tag in Fc region or Igα/β of VRC01-IgM-BCR.

**Author response image 2. respfig2:** Schematic illustration showing average FRET between ReAsH labeled at Igβ and AF647-conjugated polyclonal Fab fragment of antibody in Fc region. Top: FRET between ReAsH and AF647 (binding to IgM Fc5μ, which contains most of Cμ3/4 region) in IgM-BCR. Down: FRET between ReAsH and AF647 in IgG-BCR.

2) The authors should be able to generate CoA488 and ReAsH labeled soluble IgM and study the FRET in solution and the effect of antigen binding with absolute quantification. While this is an IgM antibody, there are ways to produce these in 293 cells without or with J-chain to generate monomers or pentamers. This would enable a systematic study of the conformational dynamics of the receptors in solution with and without ligand and binding and forced clustering. The system could be used to calibrate the FRET signature of cell surface FRET. These experiments could be done in bulk, single cell or microscopy modes.

We took this wonderful suggestion and produced soluble VRC01-IgM monomers (no J-chain) with ybbR tag inserted at N terminus and tetracysteine tag inserted in Cμ2 region in 293F cells, labeling them with CoA 488 and ReAsH (Figure 1—figure supplement 2). Since we used dequencing FRET and FLIM-FRET to examine the FRET efficiency within extracellular domain in VRC01-IgM-BCR, we also took advantage of these two methods for FRET measurement in soluble VRC01-IgM. FRET signals generated with same method in soluble IgM or IgM-BCR are comparable (Figure 2, Figure 2—figure supplement 2), indicating the FRET signal within soluble IgM could be used to calibrate the FRET signal within IgM-BCR. Data generated from both methods showed no conformational change occurred between N terminus and Cμ2 region in soluble IgM upon engagement with V51 monomer antigen presented in lipid bilayer (Figure 2—figure supplement 2). These results are consistent with previous findings (Davies et al., 1983; Davies et al.,1990; Metzger et al.,1978), in which the crystal structures of antibodies indicate that soluble antigen engagement seems incapable of propagating significant conformational changes to membrane-proximal regions of Ig. We also discussed the difference conformational change between soluble IgM and IgM-BCR in the Discussion section.

3) The authors should verify that gp120 is monomeric and could determine the consequences of gp120 binding to the monomeric or pentameric soluble IgM in terms of any evidence for generating of higher order structures, which would suggest that the gp120 is not monomeric or can be induced to form oligomers when engaged by the Fab. But the resulting multimers could be characterized by gel filtration or analytical ultracentrifugation.

We agree with the reviewer on the importance of verifying that gp120 antigen is in monomeric format. We used V51 antigen fragment, which is one epitope-focused antigenic domain recognized by VRC01 antibody and VRC01-BCR expressing B cell as reported in our recent published paper in HIV-1 vaccine studies (Wang et al., 2018). We used blue native-polyacrylamide gel electrophoresis (BN-PAGE) to determine the oligomeric states of V51 fragment (Figure 1—figure supplement 1E). We also used V51 monomer (which has only one epitope recognized by VRC01 antibody) and V51 trimer to stimulate the calcium signal of VRC01-IgM-BCR expressing Ramos B cells. In contrast to the calcium influx induced by V51 trimer, V51 monomer is unable to trigger the activation of B cells (Figure 1—figure supplement 1F).

4) Parts of Figure 1, Figure 2Figure 6 and Figure 7 could be used as a core for a revised paper that provides a better-controlled study that focuses on the gp120 system. The authors may have sufficient time to generate results with solid phase gp120, which can be attached to a solid substrate or presented in supported lipid bilayers.

We agree with the reviewers on this point. In this revised manuscript, we repeat all the FRET measurements with gp120 presented in lipid bilayers (Figure 2B), using these data to replace the data generated with NP system in the original manuscript. In brief, to present gp120 antigen in monomeric form, V51 monomer antigen fragments fused with poly-histidine tag were tethered on the surface of Ni^2+^-containing planar lipid bilayer (PLB) membranes as reported in our published paper (Zhang et al., 2013). In addition, Figure 1, Figure 2, Figure 6 and Figure 7 in the original manuscript were revised according the reviewers’ suggestions and the modified figures (Figure 1, Figure 2, Figure 4 and Figure 5 in the revised manuscript) were used as a core. Data generated with NP system were deleted in the revised manuscript.

The revisions should also address:1) Does the BCR undergo a conformational change detected in the ectodomains upon antigen binding? With soluble IgM they might hope to be able to model the FRET and even estimate average distances.

We measured the conformational change of soluble tagged VRC01-IgM according to the reviewers’ suggestion (please see Essential revision 2). The results showed that no FRET changes occurred upon membrane-bound V51 monomer antigen engagement (Figure 2, Figure 2—figure supplement 2). Since we are focusing on the conformational changes in BCR complex, it is our opinion that the relative distance change indicated by FRET measurement may reflect the conformational states, as the published papers describing the conformational change of TCR or GPCR after binding with ligand without calculating the accurate distance change (Lee et al., 2015; Vafabakhsh et al., 2015; Nuber et al., 2016).

2) Does the conformational change correlate with downstream activation signaling pathways? The experiments with IgGa/b labeling in the gp120 system may shed light on this.

We quantified the phosphorylated Igα (p-Igα) in stimulated A20II1.6 B cells and examined the association of the level of synaptic recruitment of p-Igα with FRET between N terminus and Cμ2 domain of VRC01-IgM-BCR, using phospho-CD79A (Tyr182) antibody, which has been used for quantifying the strength of B cell activation (Cell Signaling Technology, Volkmann et al., 2016) (Figure 2J), and also determined the association of p-Igα and FRET between mIg heavy chain and Igβ of VRC01-IgG-BCR (Figure 5I). In consistent with the results by measuring the phosphorylated Syk in the NP system from the original manuscript, we found inverse correlation between conformational change and the strength of B cell activation in VRC01-IgM-BCR, in which conformational change occurs in the ectodomains upon antigen engagement. Similarly, in VRC01-IgG-BCR, conformational change in the spatial relationship between mIg heavy chain and Igβ was found to be inversely correlated with the strength of B cell activation.

3) The site specific labeling of the BCR is a novel and useful tool. But to interpret the donor-dequenching data, it is important to know the labelling efficiency with the donor and acceptor. The relative stoichiometry of the labels is particularly critical for comparing the different conditions and BCR constructs. FLIM is a good way to determine the level of FRET irrespectively of labelling stoichiometry, and the authors do use it in one experiment, but unfortunately, they do not derive the information about the fraction of donor molecules in FRET and their FRET level. This could be achieved with soluble IgM and lessons applied to understanding what is happening on the cell surface.

We fully agree with the reviewers that the relative stoichiometry of the labels is critical for comparing FRET changes under different conditions (with or without antigen) or in different BCR constructs (IgM- or IgG-BCR). Although it is difficult to determine the accurate labeling efficiency of the donor and acceptor, respectively, in all the experiments, we analyzed and compared the FRET signals in subsets of cells with equal donor intensity and acceptor intensity (Figure 2D, 2I; Figure 5C, 5H; Figure 3—figure supplement 1; Figure 6—figure supplement 1). For FLIM-FRET method, we showed the fraction of donor molecules in FRET and their FRET level in this revised manuscript (Figure 2H; Figure 5G). We used FLIM-FRET to study the FRET within soluble VRC01-IgM, obtaining the comparable FRET level as VRC01-IgM-BCR on the surface of cells (Figure 2—figure supplement 2C-F). By FLIM-FRET measurement, we found that in contrast to the FRET change occurred in VRC01-IgM-BCR expressed on cell surface upon antigen engagement, such FRET change was not observed in soluble VRC01-IgM after binding with V51 monomer antigen (Figure 2—figure supplement 2).

4) Controls for the FRET photobleaching measurements using donor only and acceptor only samples were missing. These are important as there is often some bleedthrough and also some photobleaching-induced photoconversion. Given that the antigen-induced FRET changes are small, these additional corrections may be significant.

We thank the reviewers for raising this point. We used donor only and acceptor only samples (VRC01-IgM-BCR, in which conformational change occurred within ectodomains) for FRET photobleaching measurements. For FRET between N terminus and Cμ2 domain, we found almost no bleedthrough occurred in donor only or acceptor only sample (please see Author response image 3 and Table A).

For FRET between mIg heavy chain and Igβ, we found weak bleedthrough in donor only or acceptor only sample (please see Author response image 4 and Table B). We calculated the “FRET efficiency” resulted from the bleedthrough in donor only and acceptor only samples (VRC01-IgG-BCR, in which conformational change occurred between mIg heavy chain and Igβ), finding that antigen binding caused no changes on “FRET efficiency” in both samples (please see Table B and Author response image 5), indicating the FRET change upon antigen engagement in VRC01-IgG-BCR was mainly resulted from “true” FRET signals in donor + acceptor samples.

**Author response image 3. respfig3:** Representative images of donor only and acceptor only samples of tagged VRC01-IgM-BCR before and after acceptor photobleaching.

Average CoA 488 MFI Pre-photobleachAverage CoA 488 MFI After-photobleachAverage ReAsH MFI Pre-photobleachAverage ReAsH MFI After-photobleachDonor only samples146.6148.71.70.9Acceptor only samples2.93206.58.5

Table A. Bleedthrough measurement of donor only (CoA 488 labeled at N terminus of mIg heavy chain) and acceptor samples (ReAsH labeled at Cμ2 region of mIg heavy chain) of tagged VRC01-IgM-BCR

**Author response image 4. respfig4:** Representative images of donor only (ReAsH labeled in Igβ) and acceptor only (AF647 labeled in Fcγ region) samples of tagged VRC01-IgG-BCR before and after acceptor photobleaching.

Average ReAsH MFI Pre-photobleachAverage ReAsH MFI After-photobleachAverage AF647 MFI Pre-photobleachAverage AF647 MFI After-photobleachDonor only samples (Ag-)187.9204.69.51.1Acceptor only samples (Ag-)16.432.1171.411.8Donor only samples (Ag+)192.9208.513.71.5Acceptor only samples (Ag+)1731171.419

Table B. Bleedthrough measurement of donor only (ReAsH labeled in Igβ) and acceptor samples (AF647 labeled in Fcγ region) of tagged VRC01-IgG-BCR

**Author response image 5. respfig5:** “FRET efficiency” in donor only (ReAsH labeled in Igβ) and acceptor only (AF647 labeled in Fcγ region) samples of tagged VRC01-IgG-BCR upon antigen engagement. Left: donor only samples; Right: acceptor only samples.

5) Even if the authors don't perform the calibration studies with soluble IgM, they need to deconvolve intermolecular and intramolecular FRET. Its felt that soluble monomers and pentamers will provide critical insight into this, but otherwise another method to look separate out these effects is needed.

We fully agree with the reviewers that intermolecular and intramolecular FRET shall be deconvolved. To exclude potential effects from intermolecular FRET, we performed FRET experiments in 293T cells expressing dually tagged VRC01-IgM-BCR or two types of singly tagged VRC01-IgM-BCR. In detail, plasmids carrying dually tagged VRC01-IgM-BCRs were diluted with plasmids carrying untagged BCRs and expressed in 293T cells to generate only intramolecular FRET (Figure 2—figure supplement 1A), while plasmids carrying two types of singly tagged VRC01-IgM-BCRs (ybbR tag and tetracysteine tag are in different BCRs, respectively) were diluted with untagged BCRs and expressed in 293T cells to serve as a negative control presenting potential intermolecular FRET (Figure 2—figure supplement 1B). We found that antigen binding caused significant lower FRET between CoA 488 and ReAsH in cells expressing dually tagged VRC01-IgM-BCR (Figure 2—figure supplement 1C). Meanwhile, FRET signal between CoA 488 and ReAsH in singly tagged VRC01-IgM-BCRs expressing cells was very low, and no FRET change was observed between these two fluorophores upon antigen engagement (Figure 2—figure supplement 1D), suggesting the alteration of FRET between N terminus and Cμ2 domain within IgM heavy chain generated by antigen-BCR interaction was mainly attributed by the increase of intramolecular distance between these two sites, rather than the intermolecular distance changes. For the suggestion that comparing the FRET within soluble IgM monomers or pentamers, we think the intermolecular FRET measurement in soluble IgM pentamers is not well applied to the IgM-BCRs expressed on the cell surface, since the there is a significant difference between the conditions in which the FRET within soluble IgM was measured and those under which we explored the FRET within BCR complex on cell plasma membrane. For example, it is reported that extracellular conformational change-mediated oligomerization of the BCR requires the WTxxST motif in the transmembrane region of mIg (Tolar et al., 2009), which is lacked in soluble Ig, indicating that interaction between BCR complex and cell membrane might be crucial for conformational change within ectodomains of BCR upon antigen engagement. In addition, according to our previous findings, the activation of IgM-BCR is dependent on mechanical forces generated by the interaction between membrane-bound antigen and surface-expressed IgM-BCR (Wan et al., *ELife*2015). However, those mechanical forces might be absent or different in the interaction between membrane-bound antigen and soluble IgM.

For the measurement of FRET between mIg heavy chain and Igβ, it is difficult to distinguish the FRET signal within the one BCR complex or between two neighboring BCR complexes. However, considering two reasons, we think the conformational change between mIg and Igβ of VRC01-IgG-BCR observed here was mainly attributed to the FRET signal within the individual BCR complex. First, based on the findings by Tolar et al., 2005), FRET between neighboring BCR complexes increased upon antigen engagement, while in our observation, FRET between mIg and Igβ of VRC01-IgG-BCR decreased, indicating the FRET decrease was mainly explained by the distance change between mIg and Igβ in one BCR complex, rather than the distance change between mIg and Igβ in two distinct BCR complexes. Second, according to the findings in TCR (Lee et al., 2015), the contribution of FRET signal between two adjacent TCR complexes to the whole FRET signal is very low.

6) The authors should provide a better structural model for the conformational changes implied by the FRET measurements. It seems that there are limited number of ways to explain the results and authors should be able to provide schematics to systematically explore these, which may clarify experiments needed to disambiguate these in this study or the future.

We took this suggestion and presented a new model for the conformational changes indicated by the FRET changes (Figure 7).

[Editors' note: further revisions were requested after re-review, as described below.]

There is a general agreement that the FRET measurements have been improved by the revision and that conformational changes in the extracellular domain are being detected. However, all reviewers felt that the interpretation that these conformational changes are causal for BCR triggering is not demonstrated. In fact, one possibility that was raised is that the conformational change is only observed in the receptor that lacks the cytoplasmic tail of IgG, which is known to have signaling activity. Thus, one possibility not explored by you is that the conformational change results in some mode of negative feedback that is averted by signaling through the cytoplasmic tail of IgG isotype, but not IgM.

We fully agree with the editors and reviewers that it is premature to conclude that antigen-induced conformational changes are causal for BCR activation based on the observations in this study. Thus, we have modified the language throughout the revised manuscript, avoiding claiming that BCR conformational change “trigger” or “promote” B cell downstream signaling. For the physiological meanings of the observed BCR conformational changes, we have discussed the possible explanations and future study directions in the following replies to each major point.

Essential revisions:1) Doubts remain on the physiological meanings of the observed effects, especially with regard to the initiation of BCR signaling. The main conclusion of the authors is that intramolecular conformational changes in surface Ig molecules transport the information of extracellular antigen binding across the plasma membrane to induce intracellular BCR signaling (see subsection “Conformational changes were distinct in soluble IgM and IgM-BCR” and Discussion section). This main conclusion is not justified by the data presented. FRET experiments shown in Figure 3 indicate that those BCR classes with enhanced signaling potency (i.e. those with an Ig-γ tail) did NOT show any intramolecular conformational changes. Hence, if the observed conformational changes are a physiologically meaningful mechanism, it is not to "promote" or "facilitate the intracellular signaling in B cells" as suggested by the authors. In fact, it is well established that surface IgG has a greatly improved signaling capacity compared to IgM because the latter lacks a cytoplasmic signaling tail (Martin and Goodnow, 2002; Wakabayashi et al., 2002; Engels et al., 2009; Liu et al., 2010). These reports are not discussed in the present manuscript. If anything, the presented results could be interpreted as an inhibitory mechanism since the changes are specific to BCRs with reduced signaling capability (i.e. BCRs without an Ig-γ tail). Also, the data shown in Figure 2J are not really supporting the idea that conformational changes in the extracellular domains of IgM drive intracellular signaling, since it is not clear how the 'immunological synapses' of cells with high and low FRET signals look like. A reduced FRET signal could simply be an indicator for efficient antigen binding and efficient BCR clustering, which would explain why those cells have enhanced amounts of phospho-CD79.Another limitation of the study that remains is the lack of a mechanistic explanation to the main observation, which is that both intra- and intermolecular conformational changes in either surface IgM or the IgG-BCR complex depend on the absence or presence of an intracellular Ig-γ tail. How can the transfer of the IgG tail to IgM prevent a conformational change between Fab and Cμ2 domain? An inside out signaling mechanism can be excluded, since this phenomenon is also observed in 293 cells. The illustrations shown in Figure 7 and the Discussion section do not provide any clues and in combination with the unclear situation regarding the physiological meaning of the observed effects, the impact of the entire study on the field remains limited.

We agree with the editors and the reviewers that physiological meanings of conformational changes within BCR observed in this study need more investigations, thus we should not conclude in this manuscript that conformational changes are causal for BCR-induced B cell activation. As mentioned above, we have modified our description throughout the revised manuscript to avoid drawing these types of conclusions. Moreover, we discussed in the revised manuscript that the actual role (activating or inhibitory effect) of the reported BCR conformational changes in B cell activation can be complex depending on the context of the antigens, the concept of which can be reflected from the investigations on conformational changes of other membrane receptors, such as GPCR. Agonist and antagonist can induce distinct conformational changes of GPCR, indicating that different conformational changes within GPCR can trigger different physiological functions (Olofsson et al., 2014; Vafabakhsh et al., 2015). As for one specific comment from the editors and the reviewers, the observation that conformational change within mIg of IgM-BCR (which was found to be positively associated with the B cell activation level in this study) did not occur in the presence of IgG cytoplamsmic tail seems to contradict with the previous findings, which reported the enhancement of IgG cytoplamsmic tail on the antigen-induced B cell activation (Wakabayashi et al., 2002; Liu et al., 2010; Chen et al., 201). We would like to propose the following two possible explanations, both of which have been included into the Discussion section of the revised manuscript:

One possible explanation is that spatial distance change between N terminus and Cμ2 domain of mIg in IgM-BCR could be interpreted as an inhibitory mechanism, as proposed by the reviewers. Antigen-induced conformational change between N terminus and Cμ2 domain might inhibit further BCR activation through an unexplored negative feedback mechanism (Please see Figure A as below in this document). An interesting hypothesis is that distance change between N terminus and Cμ2 domain might affect the BCR-antigen bond lifetime. As accumulation of bond lifetimes of TCR-pMHC is required for T cell signaling according to a published study (Liu et al.,2014), it shall be intriguing to explore the effect of antigen-induced conformational change on BCR-antigen bond lifetime. This effect could be potentially investigated by combining biomembrane force probe (BFP) technology in the literature and BCR site-specific labeling methodology developed in this study. Since BFP is a highly tech-demanding system, we are going to collaborate with professor Jack Wei Chen in Zhejiang University (please refer to the following link: https://person.zju.edu.cn/en/jackweichen) for this investigation.

**Author response image 6. respfig6:** Potential inhibitory effect of conformational change between N terminus and Cμ2 domain of mIg in IgM-BCR.

Another possible explanation is that conformational change in the spatial relationship between N terminus and Cμ2 (or Cγ2) domain is only required for IgM-BCR (or IgG-BCR equipped with IgM cytoplasmic tail) to transmit antigen-binding signal to membrane-proximal domain, inducing an orientation to expose a clustering interface on the Cμ4 (or Cγ3) domain (Tolar et al.,2009), facilitating BCR triggering (Please see Figure B as below in this document). In contrast, this conformational change is probably not required for IgG-BCR (or IgM-BCR equipped with IgG cytoplasmic tail) to transmit the extracellular signal of antigen-binding to induce the potential conformational change near the Cγ3 (or Cμ4) domain. This hypothesis seems to be consistent with the previous findings that IgG cytoplasmic tail lowers the threshold of mechanical force to induce IgG-BCR activation (Wan et al., 2015; Wan et al.,2018). Considering that the mechanical force might be closely related with the conformational change, it is reasonable to speculate that IgG cytoplasmic tail might lower the threshold of B cell activation through bypassing the required conformational change between N terminus and Cμ2 domain within mIg in IgM-BCR. Thus, it will be constructive to investigate the association between mechanical force and conformational change of BCR in the future study. Combing TGT-NP system described in our previous studies (Wan et al., 2015; Wan et al.,2018) and experimental system that has been established in this manuscript, our future studies will investigate the association between mechanical force and the antigen-induced conformational changes within extracellular domain of BCR.

**Author response image 7. respfig7:** Potential activating effect of conformational change between N terminus and Cμ2 domain of mIg in IgM-BCR.

We also agree with the reviewers that the association between antigen-induced conformational changes and B cell downstream signaling observed in Figure 2J and Figure 4I can be explained by several different ways. Due to the limitation of our experimental system, it is difficult to explore the causal relationship between conformational changes and the recruitment of signaling molecules. It is possible that decreased FRET signal could be a consequence of more efficient antigen binding of the cell, thus enhancing the recruitment of phospho-CD79A, as the reviewers have suggested. It is our opinion that the current bulky-based experimental system may not be able to explicitly address this question. We are now trying to establish a single-molecule FRET experimental system to couple the conformational changes and activation level of individual BCR molecules with equal efficient antigen binding on cell membrane. Since it is a highly tech-demanding system, we are collaborating with professor Chunlai Chen in our university (please refer to the following link: http://life.tsinghua.edu.cn/publish/smkx/11529/2018/20180519040537657202236/20180519040537657202236_.html), who is a co-author of this current manuscript, to accomplish this study.

In addition, we agree with the reviewers that mechanistic explanation to the effect of IgG cytoplasmic tail on the antigen-induced conformational changes of BCR remains to be solved. We speculate that through its interaction with cell membrane (positively charged IgG cytoplasmic tail can associate with the negatively charged acidic phospholipids in the inner leaflet of the plasma membrane, Chen et al., 2015), IgG cytoplasmic tail might alter the basal conformational state of BCR complex (in fact, we found that it can affect the distance between mIg and Igα/β in the absence of antigen, as detailed in reply to major issue #2), or alter the mechanical force sensing capability of BCR (Wan et al.,2018), to influence the antigen-induced conformational changes of BCR. We have included these discussions into the revised manuscript. To test our hypothesis, we are now trying to analyze the structure of BCR complex by cryo–electron microscopy. Structure of IgM- and IgG-BCR equipped with different cytoplasmic tails will be compared, to explore whether the cytoplasmic tail will affect the basal conformational state of BCR complex. Since it is a highly tech-demanding system, we are now collaborating with professor Xueming Li from the School of Life Sciences in our university (please refer to the following link: http://life.tsinghua.edu.cn/publish/smkx/11529/2018/20180519041527828778177/20180519041527828778177_.html) to accomplish this study.

Although the physiological meanings of the observed antigen-induced conformational changes of BCR remains to be solved, in this study, by combining site-specific labeling methodology and FRET-based assay to monitor conformational changes in the extracellular domains within BCR complex upon antigen engagement, we tried to establish new experimental platforms and provide some tentative clues for several long standing questions in antigen receptor biology: How the extracellular signal of antigen binding at the variable region of mIg is transduced to the intracellular ITAMs at the cytoplasmic domain of Igα/β within BCR complex? In detail, whether or not the conformational change occurs within the extracellular domains of BCR upon antigen binding? If yes, where is the exact location of such conformational change; is it at the region within the heavy chain of BCR mIg or between mIg and Igα/β? Whether or not the conformational change within mIg of BCR complex also occurs in soluble Ig molecule upon antigen binding? Whether or not the conformational change of BCR is correlated to the strength of B cell activation? In our opinion, answering these fundamental questions shall be crucial for the further investigation of triggering and regulating mechanisms in BCR activation and B cell function. In this manuscript, conformational changes within mIg and in the spatial relationship between mIg and Igβ were captured and quantified, and the correlation between conformational changes and the strength of BCR activation was also analyzed. Lastly, antigen-binding induced conformational change within mIg of IgM-BCR on cell membrane was not observed in the case of soluble IgM monomer antibodies. It is our opinion that these findings for the first time indicated that antigen-binding induced conformational changes may be truly dependent on the presence of the transmembrane and cytoplasmic domain of BCR and the membrane microenvironment, both of which are not available for soluble antibodies. These results may explain why the published crystal structural studies of antigen-antibody complex indicated that antigen binding does not transmit conformational changes to the membrane-proximal regions of Ig molecule (Metzger, 1974; Metzger, 1992; Davies and Metzger, 1983; Davies et al., 1990). To further compare the structural difference between soluble Ig and BCR on cell membrane, we are now trying to analyze the structure of BCR complex by cryo–electron microscopy with our collaborators as mentioned above.

Thus, all these results in this study may provide molecular explanations for fundamental aspects of BCR activation during transmembrane signaling transduction of B cells. Moreover, experimental systems developed in the present study (i.e. site-specific labeling of BCR and FRET-based assay) also provide a framework from which to examine other ligand-induced molecular events in BCR and other immune receptors.

2) The interpretation of the changes in FRET between the N-terminus of the Ig and the Cm2 (or Cg2) is mostly with respect to the putative changes induced by antigen that propagate to the intracellular domains. However, this FRET measures distances across the Fc-Fab hinge region, so it is likely that the majority of the movement here involves some stretching or bending at the hinge as two antigens are engaged by the two Fab arms. This should be at least mentioned and included as one interpretation in the manuscript. It is quite possible to imagine that the average location of the Fab arms in IgM vs IgG is different and therefore a different effect may be observed after antigen binding. The effect of the cytosolic tail is potentially very interesting as it suggests that its presence prevents the antigen-induced change. However, in Figure 3, the data show that it is the basal levels of FRET that are changing between constructs, not the antigen-stimulated. Thus, there may be some effect of the intracellular tail on the orientation of the Fab fragments without antigen binding, not necessarily on "conformational changes" induced by antigen. Similar effects of the tail can be seen in FRET with the Igbeta in Figure 6. This again should be included in the authors' interpretation of the data. I think it is a bit premature to conclude that these conformational changes are causal in triggering signaling.

Thanks for all of these insightful comments. We fully agree that distance change between N terminus and Cμ2 region within mIg of IgM-BCR observed in this manuscript involves stretching or bending of the hinge region between Fab and Fc fragment. Thus, we have included this interpretation into the Discussion section and Figure legend of Figure 6. In fact, considering that IgD-BCR activation is regulated by its hinge region (Übelhart et al., 2015), it is worth exploring the effect of hinge region on antigen-induced conformational changes of BCR in the future studies.

According to the reviewers’ suggestion, we also investigated the effect of cytoplasmic tail on the basal spatial relationship between Fab and Fc, or between Fc and Igβ within VRC01-BCR without antigen binding. We found that the presence of IgG cytoplasmic tail significantly increased the proximity between Fc and Igβ within VRC01-BCR without antigen binding. In contrast, IgG cytoplasmic tail had no significant influence on the relative distance between N terminus and Cμ2 (or Cγ2) region within VRC01-BCR (Please see Figure 3—figure supplement 1C, D, Figure 5—figure supplement 1C, D in the revised manuscript). Thus, we agree with the reviewers that the intracellular tail might affect the interaction between mIg and Igβ even in the case of antigen free, while the intracellular tail might not influence the location of Fab arms of IgM- and IgG-BCR. We have also included these into Discussion section of the revised manuscript.

[Editors' note: further revisions were requested prior to acceptance, as described below.]

The manuscript has been improved but there are some remaining issues that need to be addressed before acceptance, as outlined below:There is a general feeling that you have shown a link between the intracellular composition of the receptor and conformation of the extracellular antigen binding component of the receptor. The mechanism by which this is happening is not clear. We would like to publish your manuscript, but wanted to provide an opportunity for you address some specific comments to present the findings with maximum clarity in light of well known tissue with assessing statistical significance based on p values and related issue of how cells are selected for analysis.

We highly appreciate the comments from the editors and the reviewers to our study. We fully agree that maximum clarity and accuracy shall be achieved before the publication of our manuscript. As described in detail below, we have tried to revise our manuscript according to all the insightful suggestions and comments as kindly provided by the editors and the reviewers. We truly hope that our revised manuscript now fulfills the requirements/standards needed for publication in *eLife*.

Essential revisions:1) The major issue is still with the two parts of the paper that asses the effect of the intracellular IgG tail on the FRET in the extracellular domain. Since there is little mechanistic understanding how this can happen, it is important to report on the observations carefully. In Figure 5, based on t-tests, the authors conclude that adding the IgG tail to IgM or removing the tail from IgG changes the FRET levels between the Fc and Ig-β in resting cells. In contrast, in Figure 3, the authors conclude that the effect of the tail on the resting FRET between the Fc and the Fab is not significant and therefore nonexistent. This, however, seem to be just a matter of a p-value cutoff for this particular test, as there is a trend for the FRET changing with p as low as 0.063 in one of the conditions. This creates an odd conclusion – the IgG tail changes neither the resting nor the antigen-induced FRET, yet it changes whether the two are statistically separable from each other. First, it needs to be disclosed if the data in the supplementary figures (Figure 3—figure supplement 1 C,D, or Figure 5—figure supplement 1C,D) are from the same experiments as in the main figures (Figure 3, Figure 5). If yes, the statistical test are not adequate. The whole experiment should be analyzed with ANOVA with corrections for multiple hypothesis testing. Even if the statistical significance is not illuminating, I think it would be good to conclude whether there is a reproducible trend for a change in the FRET levels of the individual conditions or whether simply the data are not strong enough to conclude as to why the tail changes the "response" to antigen.

Thanks for these insightful suggestions. We care very much the concerns from the reviewers and would like to fully discuss/clarify these points. “First, it needs to be disclosed if the data in the supplementary figures (Figure 3—figure supplement 1C,D, or Figure 5—figure supplement 1C,D) are from the same experiments as in the main figures (Figure 3, Figure 5)”, for this point, the short answer is no. We took the reviewers’ suggestion and investigated the effect of IgG cytoplasmic tail on the basal spatial relationship between N terminus and Cμ2 (or Cγ2) within mIg, or between mIg and Igβ of VRC01-BCR without antigen binding. When doing this comparison, we performed new experiments (Figure 3—figure supplement 1 C, D, and Figure 5—figure supplement 1C,D) with enlarged sample size for these basal FRET calculations instead of using the original data (Figure 3 and Figure 5) with relatively small sample size. Thus, the data in the original supplementary figures (Figure 3—figure supplement 1C,D, and Figure 5—figure supplement 1C,D) were not from the same experiments as in the main figures (Figure 3 and Figure 5), which would not allow us to directly use ANOVA for statistical analysis. However, it shall be noted that the basal FRET comparison results using the original data from the main figures (Figure 3 and Figure 5) as analyzed by two-tailed t-test (shown in Author response image 8) was similar as the results using the new data from the supplementary figures (Figure 3—figure supplement 1C,D, or Figure 5—figure supplement 1C,D) which were also analyzed by two-tailed t-test. The conclusion is that IgG cytoplasmic tail significantly increased the basal proximity between mIg and Igβ, while IgG cytoplasmic tail did not significantly (but a reproducible trend can be consistently observed) increase the basal distance between N terminus and Fc within mIg.

**Author response image 8. respfig8:** Basal FRET comparison in BCRs with distinct Ig cytoplasmic tail using two-tailed t-test. From left to right: IgM vs. MMG in Figure 3, IgG vs. GGM in Figure 3, IgM vs. MMG in Figure 5, IgG vs. GGM in Figure 5.

We fully agree with the reviewers on the point that “If yes, the statistical test are not adequate. The whole experiment should be analyzed with ANOVA with corrections for multiple hypothesis testing”. Since both basal and antigen-binding FRET data from the same experiments were available for the main figures (Figure 3 and Figure 5), we thus included these basal FRET data that were from the same experiments and matched to the antigen-binding FRET data into the newly revised and performed ANOVA to analyze the effect of Ig cytoplasmic tail on both basal conformational state and antigen-induced conformational change of BCR as suggested by the reviewers. The newly analyzed results using two-way ANOVA showed a significant interaction effect (when an interaction effect is present, the effect of one factor shall be dependent on the other factor) between Ig cytoplasmic tail and antigen engagement on both FRET within mIg (Figure 3—figure supplement 2) and FRET between mIg and Igβ (Figure 5—figure supplement 2). These results of two-way ANOVA indicated that Ig cytoplasmic tail significantly affected the antigen-induced conformational change within mIg, or conformational change between mIg and Igβ of BCR. In addition, by using two-way ANOVA with Sidak’s correction for multiple comparisons, we found that the presence of IgG cytoplasmic tail significantly increased the basal proximity between mIg and Igβ of BCR (Figure 5—figure supplement 2A, B), while IgG cytoplasmic tail slightly, but non-significantly increased the basal distance between N terminus and Cμ2 (or Cγ2) within mIg of BCR (Figure 3—figure supplement 2A, B). Thus, it is our opinion that results of two-way ANOVA were consistent with results of two-tailed t-test. We have included all these results and thoughts into the revised manuscript.

While the basal FRET level within mIg in BCRs with distinct Ig cytoplasmic tail showed the absence of significant difference, we consistently observed a reproducible trend that IgG cytoplasmic tail slightly increased the basal distance between N terminus and Cμ2 (or Cγ2). We included this into the revised manuscript. It is our own opinion that Ig cytoplasmic tail may affect the antigen-induced FRET change between Fab and Fc region, rather than influence the basal FRET or the FRET upon antigen-binding, although the mechanism is still to be resolved. As discussed in the revised manuscript, it will be constructive to analyze the structure of BCR complex in the future study to find out whether or not IgG cytoplasmic tail can influence the basal conformational state of mIg. It is also worthy to investigate the association between mechanical force and conformational change of BCR, since IgG tail affects the threshold of mechanical force to induce BCR activation (Wan et al., 2015; Wan et al.,2018).

2) The authors are careful in matching the fluorescent intensity of the cells analysed in the FRET experiments and they do this by selecting cells for the analysis. Presumably there is an increase in BCR fluorescence in cells interacting with antigen. It would be worth explaining in the methods exactly how the cells were selected – are antigen stimulated cells matched to resting or vice versa?

We used several strategies to match the fluorescent intensity of cells in control group and antigen group. When capturing images of resting cells, we selected cells with high fluorescent intensity (i.e. in this step, we need to match resting cells to antigen-stimulated cells). In some cases, fluorescent intensity (donor and acceptor) of acquired cells were comparable in control group and antigen group. In other cases, the fluorescent intensity (donor or acceptor) in control group was still lower than antigen group. Thus, we excluded resting cells with relatively low fluorescent intensity and antigen-stimulated cells with relatively high fluorescent intensity to select a subset of cells with comparable fluorescent intensity for further analysis. Please see Author response image 9 for detailed information (an example of matching process for Figure 2I). We also included these in the Materials and methods section of the revised manuscript. It shall be noted that we are also concerned on this selection step and tried to validate if this matching process of fluorescent intensity can affect the result of FRET comparison. First, Author response image 9 indicated that matching process did not change the result of FRET comparison. Second, besides FRET measurement based on acceptor photobleaching method, which might be influenced by the fluorescent intensity of cells, we also confirmed our conclusion by using FLIM-FRET experiment to exclude the impact of the fluorescent intensity on the conclusions (Figure 2E-H, Figure 2—figure supplement 2D-G, Figure 4D-G).

**Author response image 9. respfig9:** FRET efficiency, mean fluorescent intensity of donor and mean fluorescent intensity of acceptor before matching the fluorescent intensity of control group and antigen group were shown. Since the fluorescent intensity of donor in control group was significantly lower, a subset of cells with comparable donor fluorescent intensity were selected as shown in red square for further analysis (top). FRET efficiency, mean fluorescent intensity of donor and mean fluorescent intensity of acceptor after matching the fluorescent intensity of control group and antigen group were shown (bottom).